*Resource*

# Phosphoproteomic screening identifies Rab GTPases as novel downstream targets of PINK1

Yu-Chiang Lai[1,†], Chandana Kondapalli[1,†], Ronny Lehneck[2], James B Procter[3], Brian D Dill[1], Helen I Woodroof[1], Robert Gourlay[1], Mark Peggie[4], Thomas J Macartney[4], Olga Corti[5,6,7,8], Jean-Christophe Corvol[5,6,7,8,9,10], David G Campbell[1], Aymelt Itzen[2], Matthias Trost[1,*] & Miratul MK Muqit[1,11,**]

## Abstract

Mutations in the PTEN-induced kinase 1 (PINK1) are causative of autosomal recessive Parkinson's disease (PD). We have previously reported that PINK1 is activated by mitochondrial depolarisation and phosphorylates serine 65 ($Ser^{65}$) of the ubiquitin ligase Parkin and ubiquitin to stimulate Parkin E3 ligase activity. Here, we have employed quantitative phosphoproteomics to search for novel PINK1-dependent phosphorylation targets in HEK (human embryonic kidney) 293 cells stimulated by mitochondrial depolarisation. This led to the identification of 14,213 phosphosites from 4,499 gene products. Whilst most phosphosites were unaffected, we strikingly observed three members of a sub-family of Rab GTPases namely Rab8A, 8B and 13 that are all phosphorylated at the highly conserved residue of serine 111 ($Ser^{111}$) in response to PINK1 activation. Using phospho-specific antibodies raised against $Ser^{111}$ of each of the Rabs, we demonstrate that Rab $Ser^{111}$ phosphorylation occurs specifically in response to PINK1 activation and is abolished in HeLa PINK1 knockout cells and mutant PINK1 PD patient-derived fibroblasts stimulated by mitochondrial depolarisation. We provide evidence that Rab8A GTPase $Ser^{111}$ phosphorylation is not directly regulated by PINK1 *in vitro* and demonstrate in cells the time course of $Ser^{111}$ phosphorylation of Rab8A, 8B and 13 is markedly delayed compared to phosphorylation of Parkin at $Ser^{65}$. We further show mechanistically that phosphorylation at $Ser^{111}$ significantly impairs Rab8A activation by its cognate guanine nucleotide exchange factor (GEF), Rabin8 (by using the Ser111Glu phosphorylation mimic). These findings provide the first evidence that PINK1 is able to regulate the phosphorylation of Rab GTPases and indicate that monitoring phosphorylation of Rab8A/8B/13 at $Ser^{111}$ may represent novel biomarkers of PINK1 activity *in vivo*. Our findings also suggest that disruption of Rab GTPase-mediated signalling may represent a major mechanism in the neurodegenerative cascade of Parkinson's disease.

**Keywords** Parkinson's disease; phosphoproteomics; PINK1; Rab GTPases
**Subject Categories** Membrane & Intracellular Transport; Methods & Resources; Post-translational Modifications, Proteolysis & Proteomics
The EMBO Journal (2015) 34: 2840–2861

## Introduction

Human mutations in genes encoding the mitochondrial protein kinase, PTEN-induced kinase 1 (PINK1), and the ubiquitin E3 ligase, Parkin, are associated with autosomal recessive Parkinson's disease (PD) (Kitada *et al*, 1998; Valente *et al*, 2004). There is accumulating evidence that these enzymes operate in a common signalling pathway that regulates mitochondrial quality control (Kazlauskaite & Muqit, 2015; Koyano & Matsuda, 2015; Pickrell & Youle, 2015). Genetic analysis in *Drosophila melanogaster* revealed that PINK1 and Parkin null flies exhibit significant mitochondrial defects and that PINK1 lies genetically upstream of Parkin (Clark *et al*, 2006; Park *et al*, 2006). In mammalian cells, PINK1 is activated in response to mitochondrial depolarisation and this stimulates

1  MRC Protein Phosphorylation and Ubiquitylation Unit, College of Life Sciences, University of Dundee, Dundee, UK
2  Centre for Integrated Protein Science Munich, Department Chemistry, Technische Universität München, Garching, Germany
3  Division of Computational Biology, College of Life Sciences, University of Dundee, Dundee, UK
4  Division of Signal Transduction Therapy, College of Life Sciences, University of Dundee, Dundee, UK
5  Inserm U 1127, Paris, France
6  CNRS UMR 7225, Paris, France
7  Sorbonne Universités, UPMC Paris 06, UMR S 1127, Paris, France
8  Institut du Cerveau et de la Moelle épinière, ICM, Paris, France
9  Inserm, Centre d'Investigation Clinique (CIC), Paris, France
10 AP-HP, Département des maladies du système nerveux, Hôpital de la Pitié-Salpêtrière, Paris, France
11 College of Medicine, Dentistry & Nursing, University of Dundee, Dundee, UK
   *Corresponding author. Tel: +44 1382 386402; E-mail: m.trost@dundee.ac.uk
   **Corresponding author. Tel: +44 1382 388377; E-mail: m.muqit@dundee.ac.uk
   †These authors contributed equally to this work

the recruitment of Parkin, a cytosolic protein, to depolarised mitochondria where it ubiquitylates multiple mitochondrial substrates to trigger the removal of mitochondria by autophagy (also known as mitophagy; Narendra *et al*, 2008, 2010; Geisler *et al*, 2010; Matsuda *et al*, 2010; Vives-Bauza *et al*, 2010). We and other groups have found that upon activation, PINK1 directly phosphorylates Parkin at serine 65 (Ser[65]) within its ubiquitin-like (Ubl) domain (Kondapalli *et al*, 2012; Shiba-Fukushima *et al*, 2012) and ubiquitin at an equivalent Ser[65] residue (Kane *et al*, 2014; Kazlauskaite *et al*, 2014b, 2015; Koyano *et al*, 2014), and together, phosphorylation of both residues leads to maximal recruitment and activation of Parkin at mitochondria (Kane *et al*, 2014; Kazlauskaite *et al*, 2014a,b, 2015; Koyano *et al*, 2014; Ordureau *et al*, 2014).

The molecular interplay of PINK1 and Parkin in a common pathway fits seamlessly with clinical observations that PD patients with PINK1 and Parkin mutations have similar phenotypes (Khan *et al*, 2002). However, the existence of additional PINK1-dependent phosphorylation sites has been suggested from the analysis of rat knockout models of PINK1 and Parkin (Dave *et al*, 2014). PINK1 knockout rats exhibited progressive neurodegeneration, whereas Parkin knockout rats remained unaffected, suggesting that PINK1 may regulate additional proteins that are essential for neuronal integrity and survival in the mammalian brain (Dave *et al*, 2014). Furthermore, over recent years, several genetic interactors of PINK1 have been identified in *Drosophila* models that can rescue the loss of function phenotype of PINK1 null but not Parkin null flies (e.g. TRAP1), suggesting that PINK1 downstream signalling may in part be distinct from Parkin (Zhang *et al*, 2013).

PINK1 is imported to mitochondria where its levels are kept low due to constitutive cleavage by mitochondrial proteases (Jin *et al*, 2010; Deas *et al*, 2011; Meissner *et al*, 2011) and proteasomal degradation via the N-end rule pathway (Yamano & Youle, 2013). However, upon mitochondrial depolarisation that can be artificially induced by mitochondrial uncoupling agents such as carbonyl cyanide *m*-chlorophenyl hydrazone (CCCP), PINK1 import via the TOM40 and TIM23 complexes is blocked and PINK1 is able to escape proteolytic cleavage and accumulate at the outer mitochondrial membrane (OMM) (Narendra *et al*, 2010) where it becomes catalytically active as judged by PINK1 autophosphorylation and phosphorylation of substrates (Kondapalli *et al*, 2012; Okatsu *et al*, 2012).

Under conditions in which PINK1 is stabilised and activated in mammalian cell lines, we have exploited advances in affinity-based methods for isolation of phosphopeptides combined with quantitative mass spectrometry to undertake a systematic analysis of PINK1-dependent phosphorylation sites in membrane-enriched fractions that contain mitochondria and associated compartments (e.g. endoplasmic reticulum) in which PINK1 signalling has been implicated. This excitingly revealed a novel role for PINK1 in the regulation of Rab GTPases.

Rab GTPases play a major role in endocytic and vesicle trafficking and are critical for neuronal function (Ng & Tang, 2008). However, to date, there is little known in relation to Rab GTPases and PINK1-dependent neurodegeneration. We therefore decided to further investigate the phosphorylation of Rab GTPases in response to PINK1 activation.

Our analysis reveals that PINK1 regulates the phosphorylation of a highly conserved residue, serine 111 (Ser[111]), of a family of Rab GTPases, namely Rab8A, 8B and 13. Using phospho-specific antibodies raised against phospho-Ser[111] for each of the Rab GTPases, we demonstrate that Rab Ser[111] phosphorylation is abolished in PINK1 knockout as well as PINK1 mutant patient-derived cells, indicating that this site is absolutely dependent on PINK1. We provide evidence that PINK1 may not directly phosphorylate these Rabs and instead may regulate an intermediate kinase and/or phosphatase that targets Rab Ser[111] for phosphorylation. To obtain molecular insights into the impact of phosphorylation on Rab GTPase function, we have purified a Ser[111]-phosphomimetic of Rab8A. We demonstrate that the addition of a negative charge significantly impairs interaction with and activation by its cognate guanine exchange factor (GEF), Rabin8. Our findings provide fundamental new knowledge on the regulation of Rab GTPases by PINK1 and suggest that monitoring Rab Ser[111] phosphorylation would represent a novel biomarker of PINK1 activity *in vivo*. Furthermore, our findings suggest that Rab GTPases may represent a molecular nexus between the PINK1 signalling pathway and other PD-linked genes.

## Results

### SILAC-based PINK1 phosphoproteomic screen

We and other groups have previously reported that the Parkinson's associated PINK1 kinase becomes activated in mammalian cells upon mitochondrial depolarisation that can be induced by mitochondrial uncouplers such as CCCP (Kondapalli *et al*, 2012; Okatsu *et al*, 2012). This leads to phosphorylation of its substrates Parkin and ubiquitin at the equivalent residue Ser[65] (Kondapalli *et al*, 2012; Kane *et al*, 2014; Kazlauskaite *et al*, 2014b; Koyano *et al*, 2014; Ordureau *et al*, 2014). To identify novel PINK1-dependent phosphorylation targets, we undertook a quantitative phosphoproteomic screen using stable isotope labelling by amino acids in cell culture (SILAC). Human Flp-In T-Rex HEK293 cells stably expressing empty-FLAG vector control, kinase-inactive (KI) D384A PINK1-FLAG, or wild-type human PINK1-FLAG were grown in "light" (R0, K0), "medium" (R6, K4) and "heavy" (R10, K8) SILAC media, respectively, for a minimum of five passages (Fig 1A). Labelling efficiency was assessed and found to be > 95 % across all four biological replicates (data not shown). Cells were treated with CCCP (10 μM for 3 h) to stimulate PINK1 catalytic activity, and membrane-enriched mitochondrial containing fractions were made by ultracentrifugation and then solubilised in 1% RapiGest. Protein amounts were determined, and equivalent wild-type and kinase-inactive PINK1 expression/stabilisation by CCCP was confirmed in each replicate by immunoblotting (Fig 1B). Cell lysates from the three different conditions were combined in a 1:1:1 ratio. Protein extracts were reduced, cysteines alkylated and digested using trypsin (Fig 1A). Digested peptides from each replicate were subjected to fractionation by hydrophilic interaction liquid chromatography (HILIC; McNulty & Annan, 2008), and 15 fractions were collected per experiment (Appendix Fig S1). Each fraction was further subjected to phosphopeptide enrichment using TiO₂ spin columns before analysis by mass spectrometry (Larsen *et al*, 2005; Trost *et al*, 2009; Fig 1A). Scatter plot analysis demonstrated a high level of reproducibility across all replicates (Appendix Fig S2).

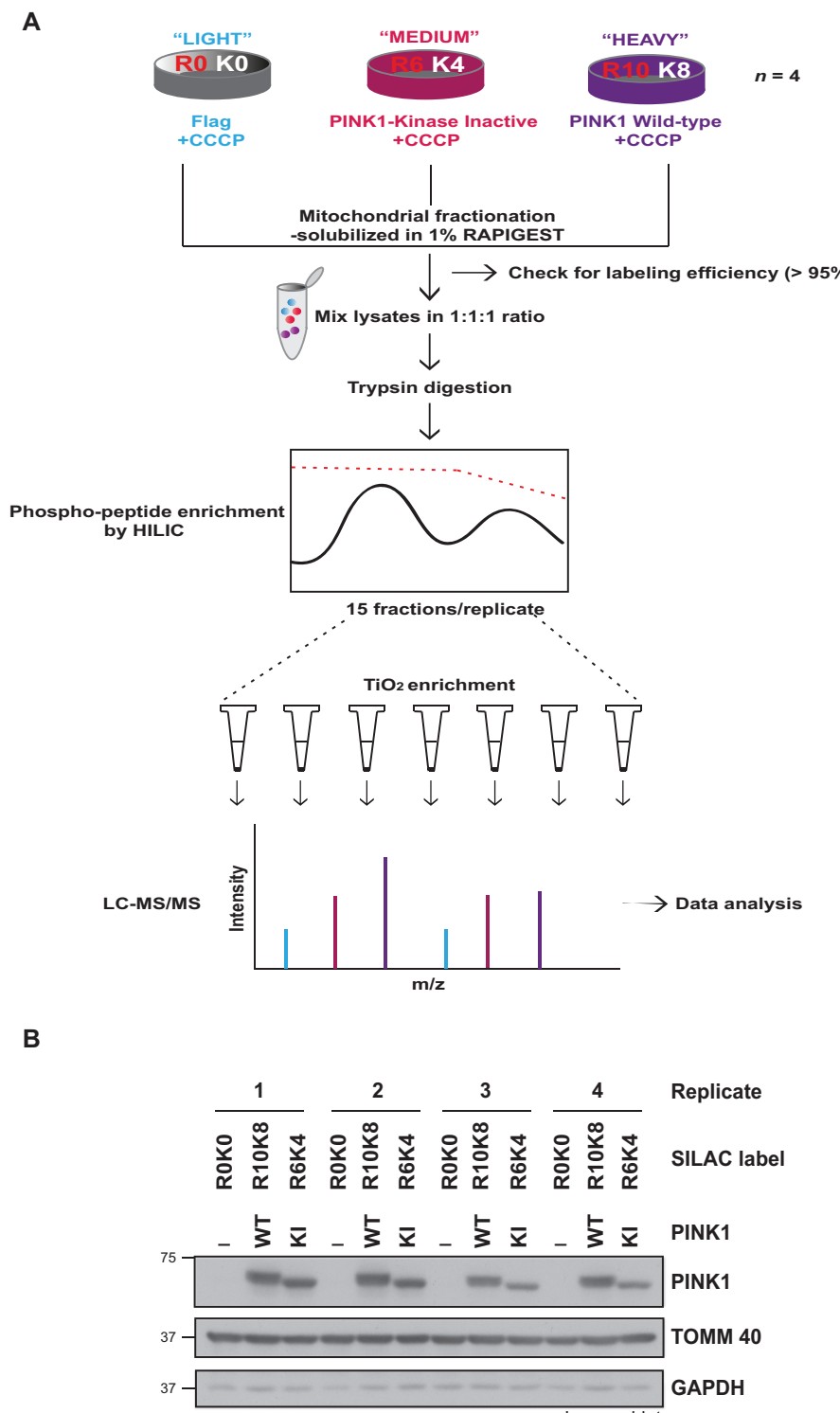

**Figure 1. SILAC phosphoproteomic approach for the identification of PINK1-dependent targets.**

A   Illustration of SILAC phosphoproteomics workflow. Flp-In T-Rex HEK293 cells stably expressing FLAG alone were cultured in unlabelled (R0K0) medium, kinase-inactive (KI; D384A) PINK1-FLAG cells were "medium" (R6K4) labelled, and wild-type (WT) PINK1-FLAG cells were "heavy" (R10K8) labelled using SILAC media containing the respective isotopes. All conditions were treated with 10 μM CCCP for 3 h and subjected to membrane fractionation. Four biological replicates of the samples were mixed in a 1:1:1 ratio, digested and fractionated by HILIC. Phosphopeptides from these fractions were enriched by TiO₂ chromatography and analysed by quantitative mass spectrometry on an Orbitrap Velos Pro mass spectrometer. Data were analysed by MaxQuant and Perseus software packages.

B   About 25 μg of membrane-enriched lysate from the mass spectrometry experiments was immunoblotted with anti-PINK1 antibody. TOMM40 and GAPDH serve as markers for mitochondria and cytoplasm, respectively.

Altogether, 52 samples were separated on a 50 cm × 75 μm online reversed-phase column and analysed on an Orbitrap Velos Pro using 4 h gradients. This led to the identification of 14,213 phospho-sites (FDR < 1%) from 4,499 gene products among which 12,374 were quantified (Table EV1). Volcano and frequency plot analyses of wild-type PINK1 versus kinase-inactive PINK1 (WT/KI; H/M) revealed that whilst most of these phosphosites remained unaffected (Fig 2A, Appendix Fig S3A), 34 phosphosites increased significantly ($P < 0.05$, > 3-fold) (Table EV2, Fig EV1A and B) and 7 phospho-sites decreased significantly (WT/KI; H/M)) ($P < 0.05$, < 0.33-fold) (Table EV3). Comparative analysis revealed 16 of these phospho-sites were also increased significantly when comparing wild-type PINK1 versus empty vector ($P < 0.05$, > 3-fold; WT/vector; H/L) (Appendix Fig S3B).

In validation of the screen, we detected a 24-fold increase, between WT and KI conditions, of a previously reported Thr257 PINK1 autophosphorylation site (VALAGEYGAVpTYR; Figs 2A and EV1C, and Table EV2; Kondapalli *et al*, 2012). We also observed a significant 17-fold increase in the ubiquitin Ser[65] phosphopeptide that we have already reported last year together with other groups as a direct PINK1 substrate (Figs 2A and EV1C, and Table EV2; Kazlauskaite *et al*, 2014b).

Strikingly among the most highly changing phosphopeptides detected, were peptides corresponding to an equivalent phosphorylation site, Ser[111], of three closely related Rab GTPases, namely Rab8A, 8B and 13 that were up-regulated 21-, 30- and 50-fold, respectively, between WT and KI conditions across all 4 replicates (Figs 2A and EV1C, Table EV2 and Appendix Fig S4). Further-more, we also detected a 6-fold increase in an equivalent Ser114/111 (Ser[114/111]) phosphopeptide of Rab1A/1B (Fig 2A, Table EV2). Multiple sequence alignment of the region surrounding Ser[111] of Rab8A, 8B and 13 revealed that these were highly conserved across all species (Fig 2B). Moreover, across the different Rab GTPases, there was strong conservation of surrounding residues with an Ala at the −1 position, an acidic Glu residue at the −3 position and Val and Glu at the + 3 and + 4 positions, respectively (Fig 2B).

### Validation that PINK1 regulates phosphorylation of Ser[111] of Rab GTPases

To confirm that PINK1 can regulate the phosphorylation of Ser[111] of Rab GTPases in cells, we over-expressed full-length human N-terminal HA-tagged Rab8A, Rab8B and Rab13 in Flp-In T-Rex HEK293 cells stably expressing wild-type human PINK1, or kinase-inactive PINK1 (Fig 3A–C). Cells were stimulated with or without CCCP for 3 h and Rab8A/8B/13 extracts immunoprecipitated with HA agarose, and phosphorylation site analysis performed by mass spectrometry. Consistent with the phosphoproteomic screen, this analysis revealed that Rab8A, 8B and 13 were phosphorylated at Ser[111] but only in cells expressing wild-type PINK1 that had been stimulated with CCCP (Fig 3A–C and Appendix Figs S5–S7). In contrast, no detectable phosphorylation of Ser[111] was detected in the absence of CCCP stimulation or in cells expressing kinase-inactive PINK1 (Fig 3A–C and Appendix Figs S5–S7).

We next raised phospho-specific antibodies that specifically recognise Rab8A, 8B and 13 phosphorylated at Ser[111] (see Materials and Methods) and assessed phosphorylation in Flp-In T-Rex

HEK293 cells stably expressing wild-type or kinase-inactive PINK1 that were transfected with HA-Rab8A, 8B or 13. Extracts were subjected to immunoprecipitation with HA agarose followed by immunoblotting with each phospho-specific antibody, and we were able to confirm that over-expressed Rab8A, 8B and 13 (Fig 3D) phosphorylations were induced, following the stimulation of wild-type but not kinase-inactive PINK1-expressing cells upon treatment with CCCP. Furthermore, mutation of Ser[111] to Ala abolished the recognition of phospho-Rab8A, 8B and 13 in CCCP-treated cells over-expressing wild-type PINK1, confirming the specificities of the antibodies generated (Fig 3D).

### PINK1 activation is essential for Rab8A, 8B and 13 Ser[111] phosphorylation in cells

We next investigated whether endogenous PINK1 is sufficient and necessary for phosphorylation of Rab8A, 8B and 13 Ser[111] in cells upon activation induced by CCCP-induced mitochondrial depolarisa-tion. Wild-type HA-Rab8A, 8B and 13 as well as the corresponding non-phosphorylatable S111A mutant of each Rab GTPase was expressed in both wild-type and PINK1 knockout HeLa cells gener-ated by CRISPR/Cas9 technology (Narendra *et al*, 2013). Cells were treated with 10 μM CCCP or DMSO for 20 h, and lysates were subjected to HA agarose immunoprecipitation followed by immunoblotting with the anti-phospho-Rab Ser[111] antibody. We observed phosphorylation of wild-type but not S111A HA-Rab8A, HA-Rab8B and HA-Rab13 in cells stimulated with CCCP, and impor-tantly, this phosphorylation was abolished in the PINK1 knockout cells (Fig 4A). Furthermore, to demonstrate that the loss of Rab Ser[111] phosphorylation was specifically due to PINK1 knockout and not an off-target effect of CRISPR/Cas9 generation, we re-expressed wild-type PINK1-3xFLAG or kinase-inactive (D384A) PINK1 into the knockout cells and we observed rescue of Rab8A, 8B and 13 Ser[111] phosphorylation only upon expression of wild-type but not kinase-inactive PINK1 (Fig 4A).

To further validate the physiological regulation of Rab GTPase Ser[111] by PINK1, we next determined whether endogenous PINK1, upon activation, was capable of phosphorylating endogenous levels of Rab8A. Wild-type and PINK1 knockout HeLa cells were stimulated with CCCP, and lysates were subjected to immuno-precipitation using a Rab8A antibody followed by immunoblotting of immunoprecipitates with either anti-phospho-Rab Ser[111] or total Rab8A antibodies. This revealed that endogenous Rab8A Ser[111] phosphorylation occurred in wild-type but not in PINK1 knockout Hela cells stimulated with CCCP to induce PINK1 activation (Fig 4B). Furthermore, phosphorylation of Rab8A was rescued by re-expression of wild-type but not kinase-inactive PINK1 (Fig 4B). We also obtained similar results in HEK293 cells (Appendix Fig S8). These results confirm that endogenous PINK1 can regulate endoge-nous Rab8A Ser[111] phosphorylation in cells.

### Rab8A Ser[111] phosphorylation is abolished in human PINK1 patient-derived fibroblasts and mouse PINK1 knockout fibroblasts

To explore the physiological relevance of PINK1-dependent Rab GTPase Ser[111] phosphorylation to Parkinson's disease (PD), we next analysed primary human fibroblasts derived from a patient with PD

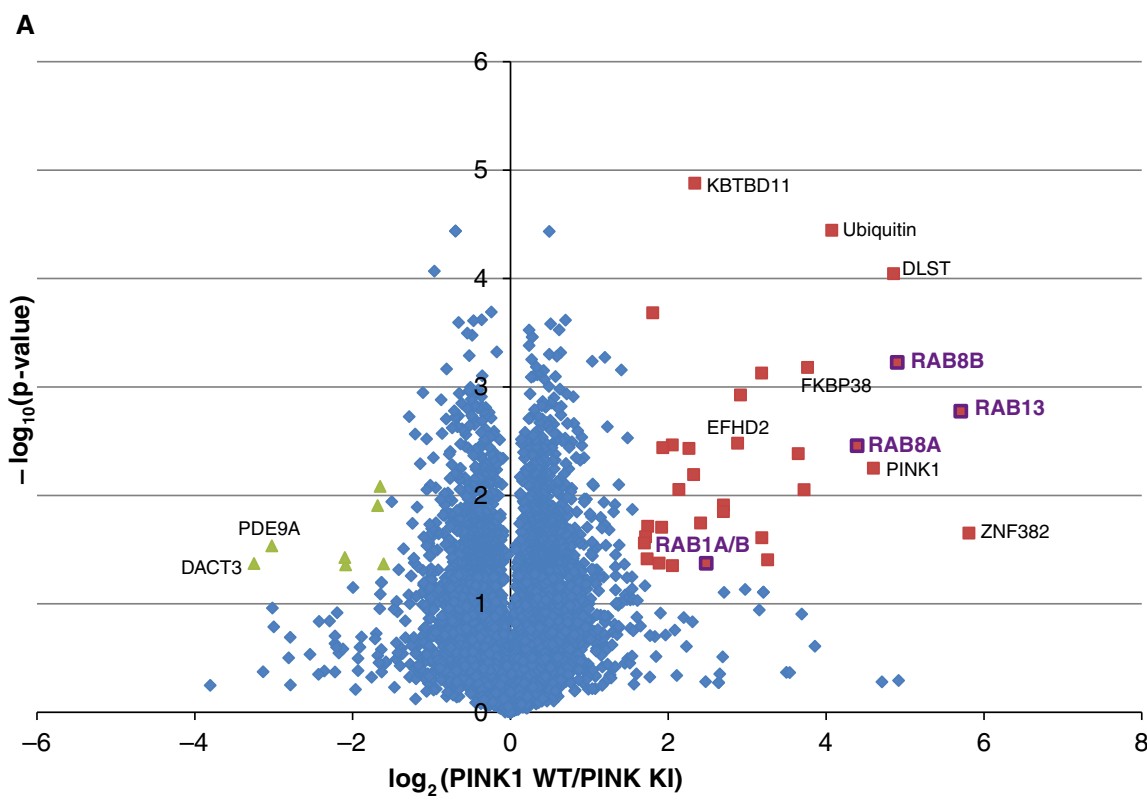

**Figure 2. Analysis of PINK1-regulated phosphoproteome and identification of Ser[111] phosphopeptides of Rab GTPases.**

A  Volcano plot highlighting significantly ($P < 0.05$, > three-fold change) up-regulated (red) and down-regulated (green) phosphopeptides identified in each screen. Rab GTPases are marked in purple.

B  Sequence alignment of Ser[111] phosphorylation site in Rab8A, Rab8B and Rab13 orthologs from mammals to *Drosophila* shows high conservation around the Ser[111] phosphorylation site (blue asterisk).

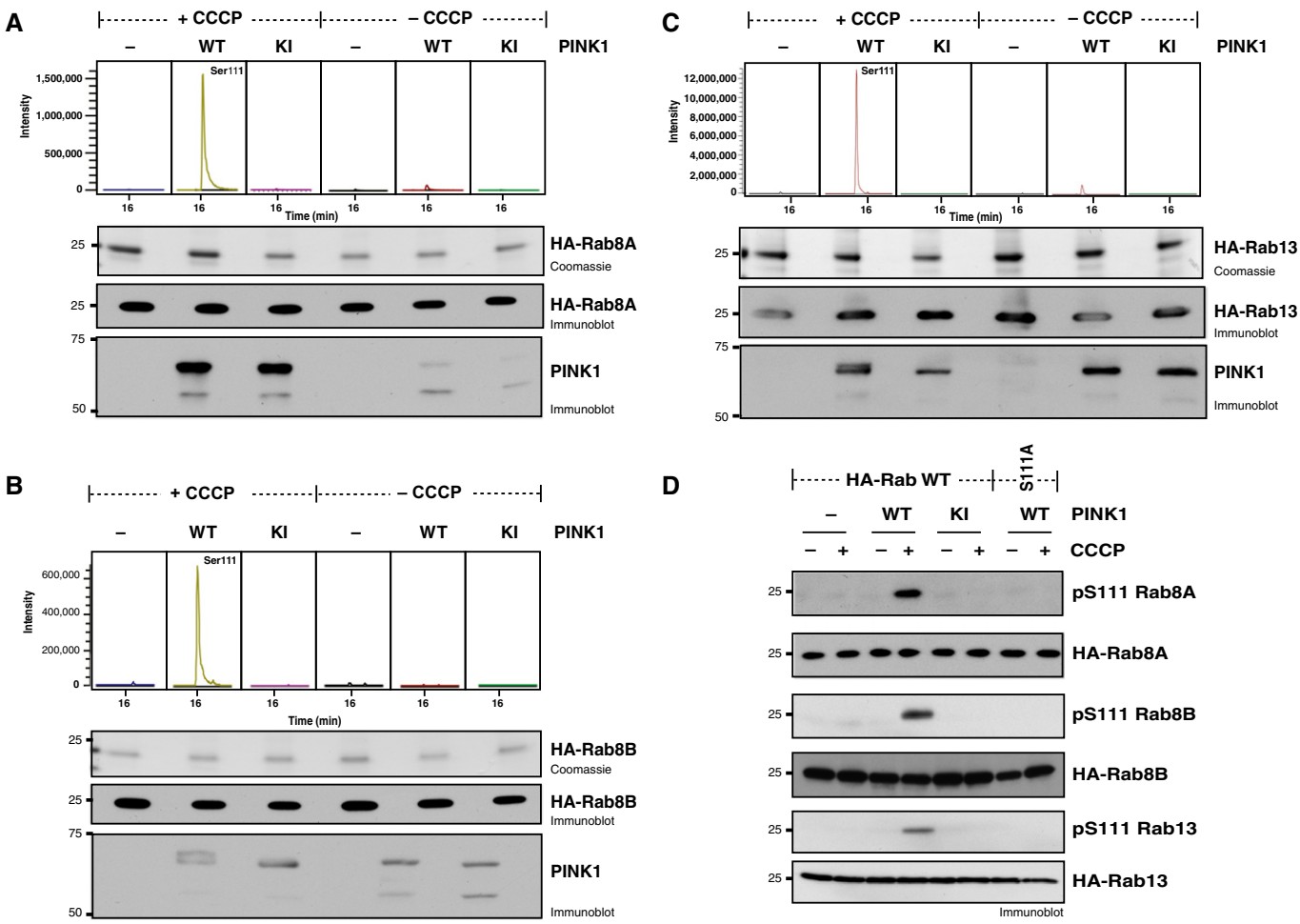

**Figure 3.  Rab8A, Rab8B and Rab13 Ser[111] phosphorylations are regulated by PINK1 upon CCCP treatment.**

A–C  Confirmation by mass spectrometry that Rab8A (A), Rab8B (B) and Rab13 (C) Ser[111] is phosphorylated upon PINK1 activation after CCCP treatment. Flp-In T-Rex HEK293 cells expressing empty-FLAG, WT PINK1-FLAG and KI (D384A) PINK1-FLAG were transfected either with HA-Rab8A (A), HA-Rab8B (B) or HA-Rab13 (C) induced with doxycycline and stimulated with 10 μM of CCCP for 3 h. Whole-cell lysates (10 mg) were immunoprecipitated with anti-HA agarose, resolved by SDS–PAGE and stained with colloidal Coomassie blue (second panel). Coomassie-stained bands migrating with expected molecular mass of HA-Rabs were excised, in-gel digested with trypsin and subjected to high-performance liquid chromatography with LC-MS/MS on an LTQ-Orbitrap mass spectrometer. Upper panel shows the extracted ion chromatogram (XIC) analysis of Ser[111]-containing phosphopeptides (8A, NIEEHApSADVEK; 8B, NIEEHApSSDVER; 13, SIKENApSAGVER)  with the combined signal intensity of the 2[+] and 3[+] forms of the peptide indicated on the y-axis. Note that the Ser[111] phosphopeptide was only detected in samples from WT PINK1-FLAG-expressing cells following CCCP stimulation.

D      Characterisation of Rab8A, Rab8B and Rab13 phospho-Ser[111] antibodies. Flp-In T-Rex HEK293 cells expressing empty-FLAG, WT PINK1-FLAG and KI (D384A) PINK1-FLAG were transfected with either WT or Ser111Ala-mutant (S111A) HA-Rab8A, HA-Rab8B or HA-Rab13, induced with doxycycline and stimulated with 10 μM of CCCP for 3 h. Whole-cell lysates (0.25 mg) were immunoprecipitated with anti-HA agarose and immunoblotted with Rab8A, Rab8B or Rab13 phospho-Ser[111] antibodies. Part of the immunoprecipitates was used to immunoblot for HA antibody as loading controls.

bearing the homozygous Q456X mutation and an unaffected individual from the same family (see Materials and Methods). Using recombinant insect PINK1 *in vitro* kinase assays, we have previously demonstrated that the Q456X mutation completely abolishes the catalytic activity of PINK1 via truncation of the C-terminal region that is essential for kinase function (Woodroof *et al*, 2011). After stimulation of fibroblasts with CCCP, lysates were subjected to Rab8A immunoprecipitation followed by immunoblotting of immunoprecipitates with either anti-phospho-Rab Ser[111] or total Rab8A antibodies. We observed total loss of Rab8A Ser[111] phosphorylation in PINK1 Q456X fibroblasts following CCCP treatment (Fig 5A). In comparison, we detected robust phosphorylation of

Rab8A Ser[111] in control human fibroblasts associated with stabilisation of full-length PINK1 after CCCP treatment (Fig 5A).

Since the Rab8A Ser[111] site and surrounding residues are highly conserved between human and mouse (Fig 2B), we next investigated Rab8A Ser[111] phosphorylation in a PINK1 knockout mouse model (Gandhi *et al*, 2009). Mouse embryonic fibroblasts (MEFs) were generated from PINK1 knockout or wild-type littermate control mice (see Materials and Methods) and stimulated with CCCP. Immunoprecipitation–immunoblot analysis revealed Rab8A Ser[111] phosphorylation in wild-type but not in PINK1 knockout MEFs after stimulation with CCCP (Fig 5B), consistent with our analysis in human PINK1 knockout HeLa cells (Fig 4B).

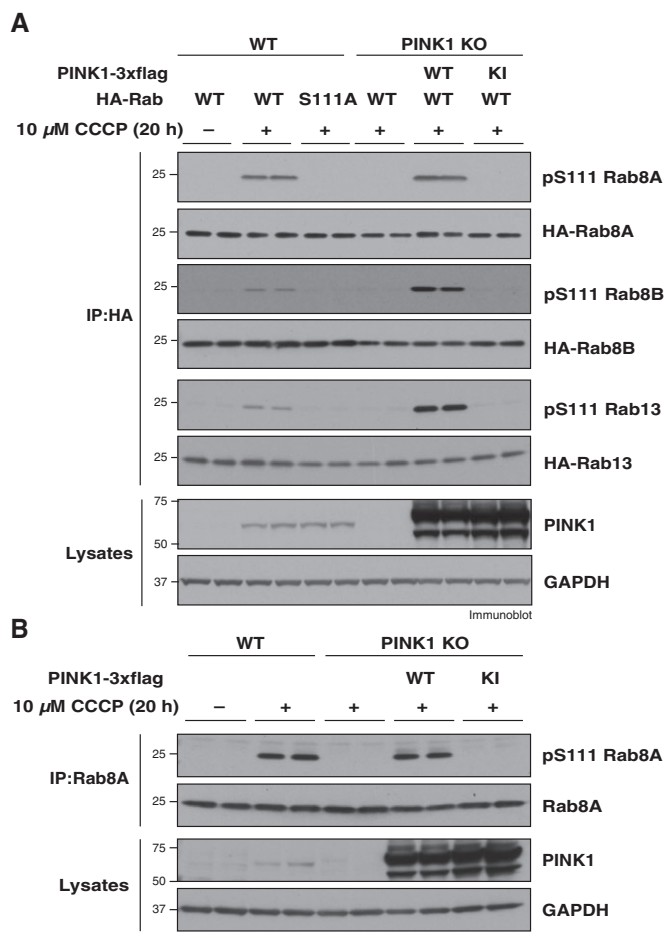

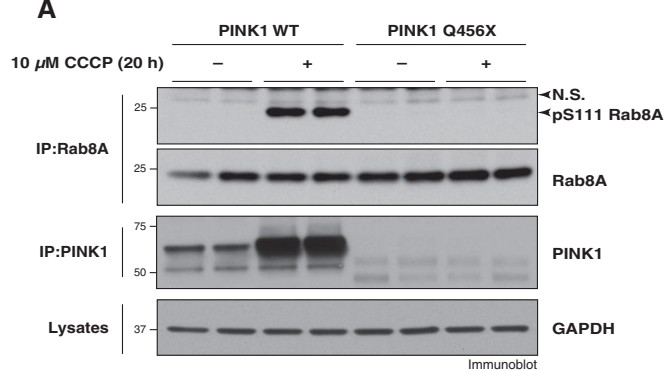

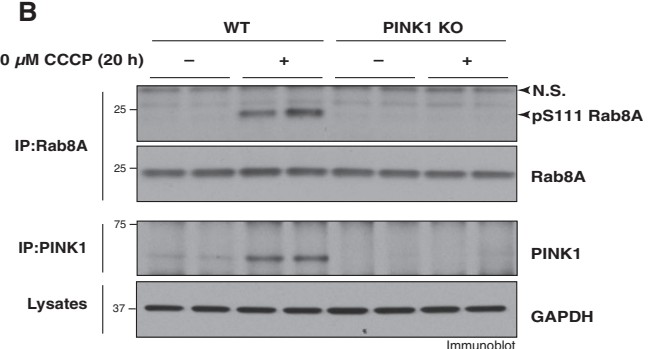

**Figure 4.    Endogenous PINK1 regulates Rab8A, Rab8B and Rab13 Ser[111] phosphorylation.**

A    PINK1 is essential for CCCP-mediated Rab8A Ser[111] phosphorylation. WT and PINK1 KO HeLa cells were transfected with either WT or Ser111Ala (S111A) mutant constructs of HA-Rab8A, HA-Rab8B or HA-Rab13. Some PINK1 KO HeLa cells were reintroduced with PINK1 by transfection of WT PINK1-3xFLAG or KI (D384A) PINK1-3xFLAG as indicated. After transfection for at least 24 h, cells were treated with DMSO as a vehicle control or CCCP for 20 h. Whole-cell lysates (1 mg) were immunoprecipitated with anti-HA agarose and immunoblotted with Rab8A, Rab8B or Rab13 phospho-Ser[111] antibody. Part of the immunoprecipitates was used to immunoblot for HA antibody as loading controls. For the lower panel, whole-cell lysates (30 μg) were immunoblotted with total PINK1 antibody to confirm PINK1 expression and with GAPDH as loading controls.

B    Endogenous Rab8A Ser[111] phosphorylation is PINK1 dependent. WT and PINK1 KO HeLa cells were treated with DMSO as a vehicle control or CCCP for 20 h. Some PINK1 KO HeLa cells were reintroduced with PINK1 by transfection of WT PINK1-3xFLAG or KI (D384A) PINK1-3xFLAG as indicated for at least 24 h before CCCP treatment. Whole-cell lysates (1 mg) were immunoprecipitated with anti-Rab8A pre-bound with protein A agarose followed by immunoblot with Rab8A phospho-Ser[111] antibody. Part of the immunoprecipitates was used to immunoblot with anti-total Rab8A antibody as loading controls.

**Figure 5.    Rab8A Ser[111] phosphorylation is abolished in Parkinson's disease patient PINK1 fibroblasts and PINK1 knockout mouse embryonic fibroblasts (MEFs).**

A    Absence of Rab8A Ser[111] phosphorylation in human mutant PINK1 patient fibroblasts. Primary skin fibroblasts were derived from a patient with homozygous PINK1 Q456X mutation or unaffected control. Cells were incubated with DMSO or CCCP for 20 h, and whole-cell lysates (1 mg) were immunoprecipitated with anti-Rab8A antibody conjugated to protein A agarose and immunoblotted with total or Rab8A phospho-Ser[111] antibody. Lysates (1 mg) were also subjected to immunoprecipitation with polyclonal anti-PINK1 antibody and immunoblotted with monoclonal PINK1 antibody. Equal loading of protein extracts was confirmed by GAPDH.

B    Absence of Rab8A Ser[111] phosphorylation in PINK1 knockout MEFs. MEFs were derived from PINK1 knockout embryos or wild-type controls (see Materials and Methods). Cells were incubated with DMSO or CCCP for 20 h, and whole-cell lysates (1 mg) were immunoprecipitated with anti-Rab8A antibody conjugated to protein A agarose and immunoblotted with total or Rab8A phospho-Ser[111] antibody. Lysates (1 mg) were also subjected to immunoprecipitation with a polyclonal anti-mouse-specific PINK1 antibody and immunoblotted with a different anti-mouse-specific PINK1 antibody. Equal loading of protein extracts was confirmed by GAPDH.

Parkin is not required for PINK1-targeting of Rab8A (Fig 4A and B). We further confirmed this in MEFs derived from a Parkin knockout mouse model (Itier *et al*, 2003) (see Materials and Methods). Immunoprecipitation–immunoblot analysis revealed Rab8A Ser[111] phosphorylation in both wild-type and Parkin knockout MEFs after stimulation with CCCP (Appendix Fig S9A and B).

Conversely to investigate whether PINK1-dependent targeting and activation of Parkin was dependent on Rab8A, we over-expressed full-length wild-type (WT) or catalytically inactive Cys431Phe (C431F) Parkin in wild-type HeLa cells or Rab8A knock-out HeLa cells generated by CRISPR/Cas9 technology (see Materials and Methods; Fig 6). Cells were treated with or without CCCP for

## Rab8A is not required for PINK1-dependent activation of Parkin E3 ligase activity

The demonstration that PINK1 can phosphorylate Rab8A Ser[111] in wild-type HeLa cells (that lack detectable Parkin) suggests that

6 h—conditions that induce stabilisation and activation of PINK1 and Rab8A phosphorylation (Fig 6). We assessed ubiquitylation of two previously reported Parkin substrates, CISD1 and mitofusin 2 (Sarraf *et al*, 2013), by immunoblotting mitochondrial extracts enriched for ubiquitylated proteins using immobilised haloalkane dehalogenase (HALO)-tagged UBA[UBQLN1] technology which preferentially binds all types of poly-ubiquitin chains (Fig 6). In wild-type HeLa cells, expressing WT but not C431F Parkin, we observed multi-monoubiquitylation of CISD1 after CCCP treatment indicative of Parkin activation and this was unaffected in Rab8A knockout cells (Fig 6). Similarly, we did not observe any difference in mitofusin 2 ubiquitylation between wild-type and Rab8A knockout cells (Fig 6). Interestingly, we observed residual ubiquitylation of mitofusin 2 in HeLa cells expressing C431F Parkin upon CCCP stimulation, suggesting that additional E3 ligases may be activated by CCCP and contribute to mitofusin 2 ubiquitylation. In future work, it will be interesting to determine whether this Parkin-independent E3 ligase activity is PINK1 dependent.

## Evidence that PINK1 does not directly phosphorylate Rab8A Ser[111]

We next investigated whether PINK1 could directly phosphorylate Rabs at Ser[111]. Sequence alignment of the Rab8A, 8B and 13 Ser[111] site and Rab1A/B Ser[114/111] with the phosphorylatable residue Ser[65] of Parkin and ubiquitin did not reveal significant sequence similarity (Fig EV2A). It has recently been suggested that the structural fold rather than the sequence may be the determining factor for PINK1 recognition of direct substrates (Wauer *et al*, 2015). Consistent with this, we have found that PINK1 is unable to phosphorylate a peptide bearing Ser[65] of the Parkin Ubl domain (Fig EV2B).

We therefore undertook a comparative analysis of structural data available on the location of the phosphorylatable residues: ubiquitin Ser[65], Parkin Ser[65], Rab8A Ser[111] and the paralogous Rab1A Ser[114]. Inspection of their structural environment (Fig 7A) demonstrates that the phosphorylated sites in ubiquitin and Parkin have a markedly different structural environment to those of Rab8A and Rab1A. In both ubiquitin and Parkin, the phosphorylated serine lies after a right-handed β-turn, before the 5th β-strand. In contrast, the phosphoserine of Rab8A and Rab1A occurs after a C-terminal helix cap and before a right-handed β-turn. The different conformations adopted by these sites suggest that distinct kinases are involved in the phosphorylation of the two groups (Fig 7A).

We next tested this prediction in phosphorylation assays of full-length untagged Rab8A with catalytically active recombinant wild-type or kinase-inactive *Tribolium castaneum* PINK1 (TcPINK1). In contrast to ubiquitin, we observed only weak phosphorylation of Rab8A by TcPINK1 with a maximal stoichiometry of approximately 0.03 moles of $^{32}$P-phosphate per mole of protein (Fig 7B). Furthermore, mutation of Ser[111] to Ala did not prevent phosphorylation of Rab8A by TcPINK1, indicating that Ser[111] is not directly phosphorylated by PINK1 (Fig 7B). To identify the sites of Rab8A phosphorylated by TcPINK1 *in vitro*, $^{32}$P-labelled Rab8A was digested with trypsin and separated by reversed-phase chromatography on a C18 column. This revealed two major $^{32}$P-labelled peptides, and a combination of solid-phase Edman sequencing and mass spectrometry revealed that each corresponded to a peptide phosphorylated at Thr74 and Thr72, respectively (Fig EV2C–E). We did not observe

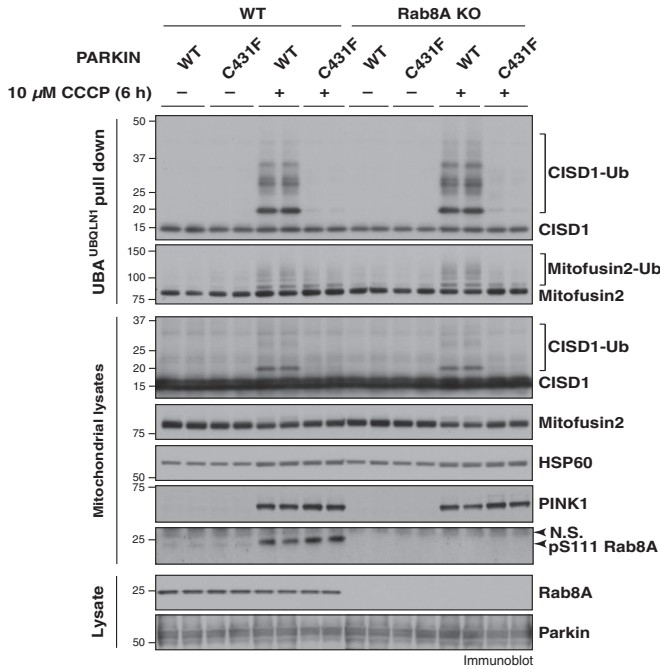

**Figure 6.  Rab8A is not required for the activation of Parkin E3 ligase activity at mitochondria in response to PINK1 activation by CCCP.**
Wild-type (WT) or Rab8A knockout (KO) HeLa cells were transfected with WT or Cys431Phe (C341F)-mutant Parkin. After transfection for 24 h, cells were treated with DMSO as a vehicle control or 10 μM CCCP for 6 h. Mitochondrial enriched extracts (mitochondrial lysate) were incubated with ubiquitin-binding resins derived from his-halo-ubiquilin1 UBA domain tetramer (UBA[UBQLN1]). Captured ubiquitylated proteins were subject to immunoblotting with anti-CISD1 and anti-mitofusin 2 antibodies. Mitochondrial lysate and total lysate were also subjected to immunoblotting with indicated antibodies for loading and protein expression controls. Phospho-Ser[111] Rab8A was detected after Rab8A immunoprecipitation from 200 μg of mitochondrial lysate with anti-Rab8A antibody.

any evidence by mass spectrometry that these Thr sites are regulated by PINK1 in cells upon stimulation with CCCP under conditions in which we do observe Ser[111] phosphorylation, suggesting that they may not be relevant *in vivo* (data not shown).

## Timecourse of Rab8A Ser[111] phosphorylation

Using Flp-In T-Rex HEK293 cells stably expressing wild-type PINK1, we have previously reported that PINK1 is activated at 5 min as judged by monitoring Parkin Ser[65] phosphorylation (Kondapalli *et al*, 2012). Under similar conditions, we next investigated the phosphorylation of Rab8A, 8B and 13 Ser[111] relative to Parkin Ser[65] phosphorylation. Cells were transfected with wild-type Rab8A, 8B, 13 or Parkin and their non-phosphorylatable Ser111Ala (S111A) or Ser65Ala (S65A) mutants, respectively. Using a phospho-specific antibody against phospho-Ser[65], we observed Parkin Ser[65] phosphorylation at 5 min as previously reported (Fig 8A) (Kondapalli *et al*, 2012). In contrast, the phosphorylation of Rab8A, 8B and 13 Ser[111] occurred significantly later at ~40 min increasing in a time-dependent fashion to 3 h (Fig 8A). The delay in Rab Ser[111] phosphorylation after PINK1 that becomes active at 5 min strongly suggests that PINK1 may not directly phosphorylate Rab GTPase

Ser[111] in cells and instead may regulate a kinase or phosphatase upstream of Rab Ser[111] consistent with our *in vitro* analysis (Fig 7B).

We next investigated the timecourse of endogenous PINK1 activation and Parkin Ser[65] and Rab Ser[111] phosphorylation in HeLa cells. HeLa cells were transfected in parallel with either wild-type Parkin or Rab8A, 8B and 13 together with their non-phosphorylatable Ser65Ala and Ser111Ala mutants, respectively. Using a phospho-specific antibody against phospho-Ser[65], we observed Parkin Ser[65] phosphorylation occurring within 10–20 min and becoming maximal at 1 h upon treatment with CCCP (Fig 8B). In contrast, under the same conditions, the phosphorylation of Rab8A, 8B and 13 Ser[111] occurred significantly later after 1 h of treatment with CCCP and increased up to 9 h (Fig 8B). Consistent with our PINK1 overexpression analysis, these results suggest that endogenous PINK1 does not directly phosphorylate Rab at Ser[111].

## Phosphorylation of Rab8A Ser[111] impairs Rabin8-catalysed GDP exchange

Rab GTPases belong to the superfamily of Ras GTPases and function as molecular switches cycling between GDP-bound inactive and GTP-bound active states (Hutagalung & Novick, 2011). To exert their function, Rabs first require to be activated in a reaction requiring guanine nucleotide exchange factors (GEFs). GEFs physiologically catalyse the release of GDP, thereby allowing Rab activation by binding of GTP, which enables interaction with effector proteins that bind with high affinity to Rabs in their GTP-bound but not GDP-bound state. We have previously structurally defined the interactions of Rab8A with its GEF Rabin8 (Guo *et al*, 2013). Rabin8 is a 460 amino acid protein that contains a central Sec2 coiled-coiled domain exhibiting GEF activity towards Rab8 (Hattula *et al*, 2002). Whilst inspection of the co-crystal structure of Rab8A and Rabin8 revealed that Ser[111] is not directly involved in the formation of the interface of Rab8A and Rabin8, the side chain of Ser[111] lies close to a negative surface patch of Rabin8 adjacent to the interaction interface (Fig 9A). We therefore hypothesised that the addition of a negative charge on Ser[111] may influence the Rab8A–Rabin8 interaction.

In view of the current challenges in chemical biology technologies to generate recombinant site-targeted phosphoproteins, we employed a Ser111Glu (S111E) phosphomimetic of Rab8A to obtain insights into the molecular consequences of Rab8A Ser[111] phosphorylation. Using a previously described homologous co-chaperone expression system (Bleimling *et al*, 2009), we expressed and purified wild-type, S111E and S111A versions of Rab8A to homogeneity (Appendix Fig S10A). Thermal shift assay analysis revealed close to identical melting points for wild-type (57.9°C for 1 and 10 μg), S111E (58.0°C for 1 μg and 58.5°C for 10 μg) and S111A (57.9°C for 1 μg and 57.4°C for 10 μg) Rab8A, suggesting that Ser[111] mutants did not significantly impair protein stability (Appendix Fig S10B).

In order to analyse the Rab8A–Rabin8 interaction, we utilised a Rabin8 catalytic assay to quantitatively determine the catalytic efficiencies ($k_{cat}/K_M$) of Rabin8-stimulated nucleotide release from wild-type Rab8A or the S111E phosphomimetic mutant (Guo *et al*, 2013). We preparatively loaded Rab8A with the fluorescent GDP analogue mantGDP and monitored the Rabin8-catalysed time-dependent displacement of mantGDP in the presence of excess GDP as judged by the decrease in mant fluorescence (Fig 9B). Strikingly, we found that the S111E phosphomimetic ($k_{cat}/K_M = 7.6 \times 10^3$ M$^{-1}$s$^{-1}$) but not a S111A mutant ($k_{cat}/K_M = 8.7 \times 10^4$ M$^{-1}$s$^{-1}$) led to a 13-fold decrease in GDP dissociation rate induced by Rabin8 compared to wild-type Rab8A ($k_{cat}/K_M = 1.0 \times 10^5$ M$^{-1}$s$^{-1}$) where the $k_{cat}/K_M$ is calculated by dividing the rate constant ($k_{obs}$) of the reaction by the enzyme concentration (Fig 9B).

We further investigated whether Rab8A Ser[111] phosphorylation could affect GTP hydrolysis since Ser[111] lies close to the functionally important switch II region in the tertiary structure (but not the primary structure; Fig EV3A). Using reversed-phase HPLC quantification to monitor GTP-to-GDP conversion over time, we did not observe any significant difference in the intrinsic GTP hydrolysis rate between wild-type and the S111E mutant of Rab8A ($k_{cat}$(wt) = $2.8 \times 10^{-5}$ s$^{-1}$, $k_{cat}$(S111E) = $1.9 \times 10^{-5}$ s$^{-1}$; Fig EV3B). In addition, we determined whether Rab8A Ser[111] phosphorylation impacts on the ability of active Rab8A (loaded with the non-hydrolysable GTP-analogue GppNHp) to bind with known effectors such as OCRL1 (Hou *et al*, 2011). Using analytical size-exclusion chromatography, the S111E phosphomimetic mutant was still able to form a stable complex with OCRL1$_{539-901}$ as well as wild-type Rab8A under these experimental conditions (Fig EV3C). Finally, we tested whether Rab8A Ser[111] phosphorylation may also influence the interaction with and de-activation by GTPase-activating proteins (GAPs). Since there is currently no known Rab8A-GAP that has been rigorously characterised in biochemical detail, we have exploited the known Rab promiscuity of the TBC domain of RabGAPs (Frasa *et al*, 2012). We have therefore expressed a TBC domain containing fragment of the Rab1-GAP TBC1D20 (residues 1–305; TBC1D20$_{1-305}$) and confirmed that it possesses Rab8A-GAP activity *in vitro* (Fig EV3D) (Sklan *et al*, 2007). Similarly, the S111E phosphomimetic mutants exhibited TBC1D20$_{1-305}$-stimulated GTP hydrolysis indistinguishable from wild-type Rab8A (Fig EV3D). This suggests that Ser[111] phosphorylation of Rab8A may not lead to a disruption in Rab8A:GAP interaction as that observed for Rab8A:GEF interaction (Fig 9B).

Overall, our analysis has revealed that phosphorylation of Rab8A at Ser[111] critically affects Rabin8 catalysis *in vitro* that would be predicted to impair Rab8A activation. In future work, it will be critical to confirm these findings using preparative phosphorylated Rab8A once the identity of the upstream kinase is elucidated or alternatively using recently described orthogonal aminoacyl-tRNA synthetase and tRNA pairs to direct incorporation of phosphoserine into recombinant Rab GTPase proteins (Rogerson *et al*, 2015).

## Evidence that Rab8A Ser[111] phosphorylation disrupts Rabin8 interaction in cells

We next addressed whether phosphorylation of Rab8A at Ser[111] influenced the interaction of Rab8A and Rabin8 in cells. We expressed wild-type (WT) HA-Rab8A, a phosphomimetic S111E mutant and a S111A mutant of HA-Rab8A in HeLa Rab8A knockout cells. Lysates were subjected to immunoprecipitation using HA agarose followed by immunoblotting of immunoprecipitates with anti-Rabin8 antibody, and we observed co-immunoprecipitation of endogenous Rabin8 with WT HA-Rab8A (Fig 9C). In contrast, we observed a drastic reduction in binding of Rabin8 with S111E but

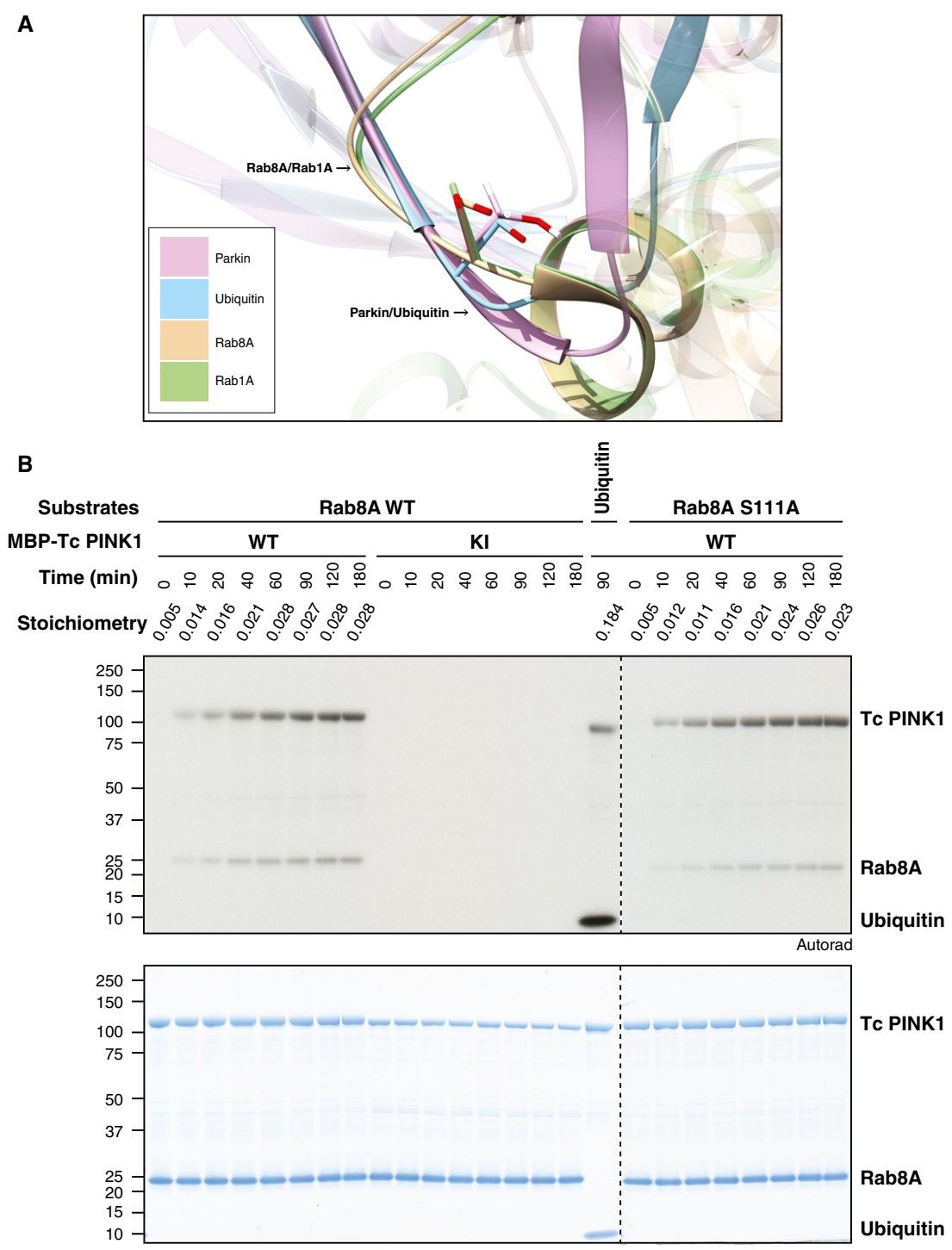

**Figure 7.  Evidence that Rab8A Ser[111] is not a direct substrate of PINK1.**

A   Ubiquitin, Parkin, Rab8A and Rab1A phosphosites adopt two distinct conformations. Ser[65] in ubiquitin (blue) and Parkin (magenta) follows a β-turn, before the start of the 5[th] β-strand. Conversely, Ser[114] in Rab1A (green) and Ser[111] in Rab8A (ochre) lie after a C-terminal helix cap just before the start of a β-turn. Representative three-dimensional structures are superimposed by C-α positions for the observed phosphosites and their sequence neighbours. Side chains for observed sites are shown as sticks, and ribbons depict backbone and secondary structure. To highlight the local environment, regions more than 3 amino acids away from the phosphosites are made transparent (see Materials and Methods for PDB IDs).

B   *In vitro* phosphorylation analysis of Rab8A by PINK1. WT or S111A-mutant Rab8A (1.2 µg) was incubated in the presence of MBP-fused WT or KI (D359A) TcPINK1 (1.1 µg) and $Mg^{2+}$-[γ-$^{32}$P] ATP for the indicated time. Samples were subjected to SDS–PAGE, and proteins were detected by colloidal Coomassie blue staining (lower panel). The [γ-$^{32}$P] incorporation to substrate was detected by autoradiography (upper panel). Cerenkov counting was used to calculate the stoichiometry of substrate phosphorylation as mol of [γ-$^{32}$P] incorporation/mol of substrate. Ubiquitin was used as a positive control of the TcPINK1 substrate.

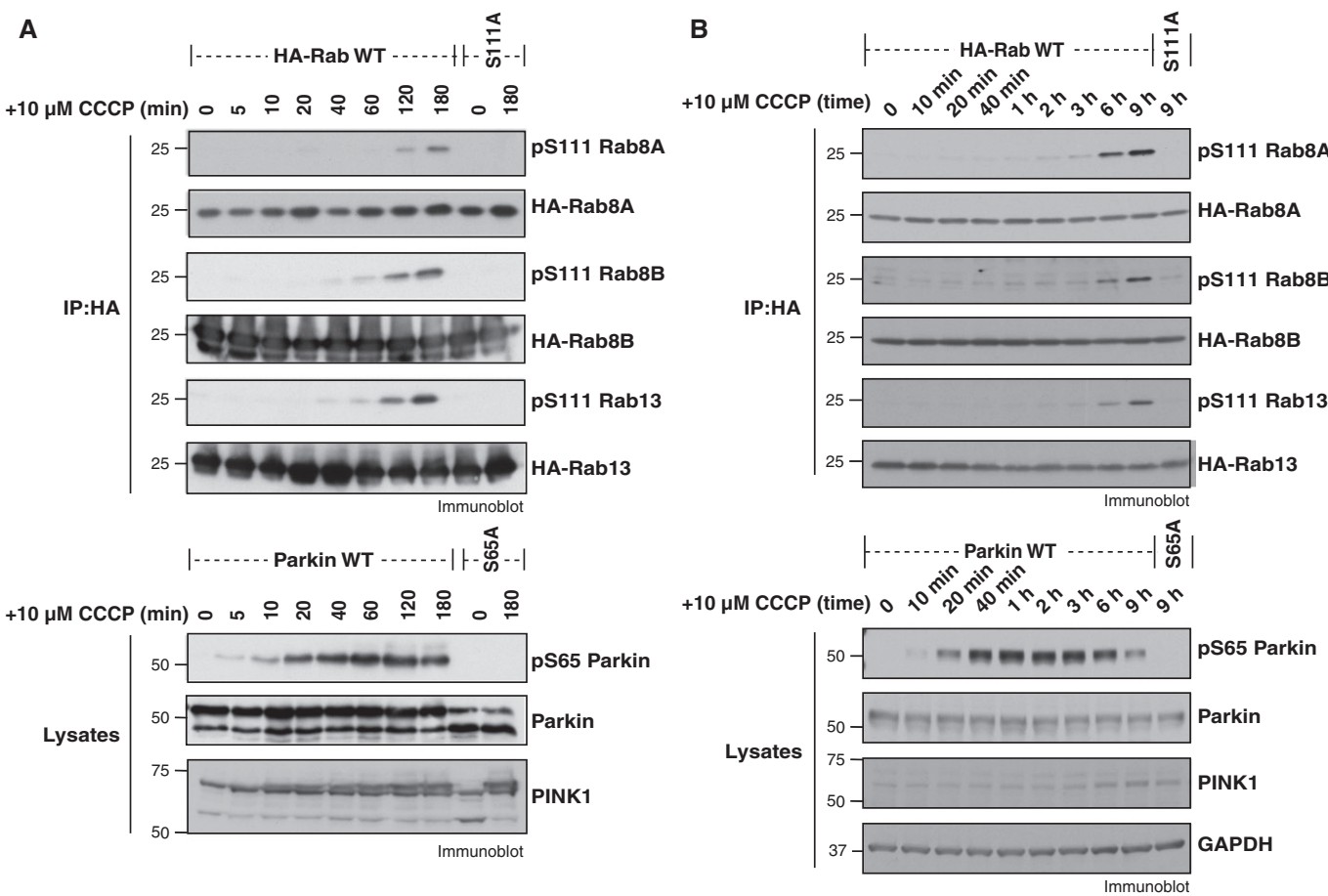

**Figure 8.  Time-course analysis of Rab8A, Rab8B and Rab13 Ser[111] phosphorylation.**

A  Time-course comparison of PINK1-mediated Rab8A, Rab8B and Rab13 Ser[111] phosphorylation vs. Parkin Ser[65] phosphorylation. Flp-In T-Rex HEK293 cells expressing WT PINK1-FLAG were transfected with either WT or Ser111Ala-(S111A) mutant HA-Rab8A, HA-Rab8B or HA-Rab13, induced with doxycycline and stimulated with CCCP for the indicated time. In parallel, Flp-In T-Rex HEK293 cells expressing WT PINK1-FLAG were transfected with either WT or Ser[65] Ala (S65A)-mutant Parkin. Whole-cell lysates (0.25 mg) were immunoprecipitated with anti-HA agarose and immunoblotted with indicated phospho-Ser[111] antibodies. Part of the immunoprecipitates was used to immunoblot for HA antibody as loading controls. For the lower panel, whole-cell lysates (30 µg) were immunoblotted with indicated antibodies.

B  Time-course comparison of endogenous PINK1-mediated Rab8A, Rab8B and Rab13 Ser[111] phosphorylation vs. Parkin Ser[65] phosphorylation. HeLa cells were transfected with either WT or S111A-mutant HA-Rab8A, HA-Rab8B or HA-Rab13 for at least 24 h before CCCP treatment for the indicated time. Whole-cell lysates (1 mg) were immunoprecipitated with anti-HA agarose and immunoblotted with indicated phospho-Ser[111] antibodies. In parallel, HeLa cells were transfected with either WT or S65A-mutant Parkin and whole-cell lysates (30 µg) were immunoblotted with indicated antibodies.

not with S111A Rab8A (Fig 9C). In parallel, co-immunoprecipitation analyses in which we co-expressed GFP-Rabin8 with either WT, S111E or S111A HA-Rab8A, we observed the converse that immunoprecipitation of Rabin8 with GFP binder sepharose resin was associated with markedly reduced binding of HA-Rab8A S111E to Rabin8 compared to WT and S111A HA-Rab8A (Fig EV4).

Overall, these cellular studies suggest that Rab8A and Rabin8 can form a complex in cells and that phosphorylation of Rab8A at Ser[111] impacts on its interaction with Rabin8 and this provides physiological relevance to our *in vitro* analysis.

## Bioinformatic analysis of Rab8A–Rabin8 surface patch interactions

The negative surface patch of Rabin8 adjacent to the Rab8A interaction interface is comprised of residues Asp203 (D203),

Glu208 (E208), Glu210 (E210), Glu211 (E211) (Guo *et al*, 2013). Given the functional relationship between Rab8A Ser[111] phosphorylation and the Rabin8 negative patch, it is tempting to speculate that this interaction may have co-evolved with PINK1. Were that the case, then for orthologues of Rab8 and Rabin8 in organisms that lack PINK1, the interaction between the charged patch and Ser[111] would not need to be conserved. To explore this hypothesis, we examined proteins orthologous to Rab8 and Rabin8 in yeast.

We first verified that yeast lacks PINK1. Examination of the entry for PINK1 in the EggNOG (Powell *et al*, 2014) orthologue database suggests PINK1 is only found in metazoans. We also employed the EggNOG hidden Markov model for PINK1 to search the NCBI NR protein sequence database with the EMBL-EBI HMMER3 server (Mistry *et al*, 2013). No significant matches were found in Saccharomyces (data not shown).

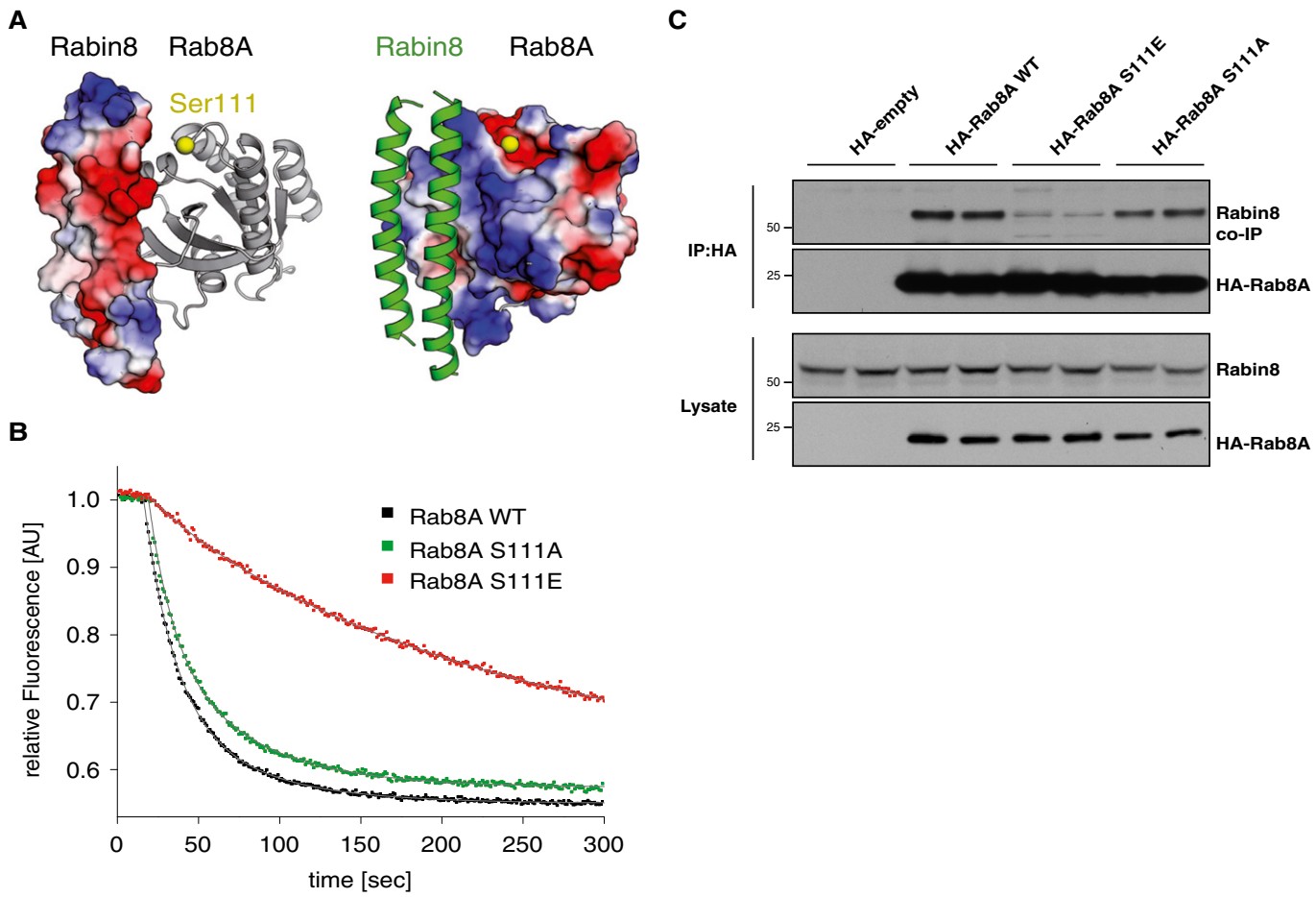

**Figure 9. Rab8A Ser[111] phosphorylation impairs Rabin8-catalysed GDP exchange *in vitro* and Rabin8 interaction in cells.**

A  Crystal structure of Rab8A in complex with guanine exchange factor (GEF) Rabin8 (Guo *et al*, 2013). Left panel: Rabin8 is shown in surface representation and Rab8A in cartoon representation. Right panel: Rabin8 is shown in cartoon representation, and Rab8A is shown in surface representation. The hydroxyl group of residue Ser[111] is shown as a yellow sphere. The surfaces are colored by their positive and negative electrostatic potentials from blue to red, respectively.

B  Rabin8-catalysed mant-GDP release from mant-GDP loaded WT, S111A and S111E mutants of Rab8A. The Rab proteins (1 μM) were incubated with 100 μM GDP in buffer (20 mM HEPES pH 7.5, 50 mM NaCl, 1 mM MgCl₂, 2 mM DTE), and the reaction was started by the addition of 0.5 μM Rabin8. The decrease in mant fluorescence was used as a measure of mantGDP release.

C  Rab8A Ser[111] phosphorylation impairs Rabin8 interaction in cells. Rab8A KO HeLa cells were transfected with wild-type (WT), S111E or S111A HA-Rab8A. Whole-cell lysates (1 mg) were immunoprecipitated with anti-HA agarose and immunoblotted with Rabin8 or anti-HA antibody. Lysates were immunoblotted with Rabin8 or anti-HA antibody to confirm equivalent expression of Rabin8 and WT and mutant HA-Rab8A in extracts.

We then compared Rab8 to Ypt1 and Sec4, its close homologues in yeast. Multiple sequence alignment of Ypt1, Sec4 and Rab8A showed that whilst these sequences align well, Rab8A Ser[111] is neither conserved in Ypt1 nor in Sec4 (Fig EV5A). The loop containing Ser[111] in Rab8A is not similar to that of Sec4, but in Ypt1, the corresponding residue is a threonine, followed by a serine. Structural data for Ypt1 and Rab8A demonstrate that they exhibit a high degree of structural homology, and it is probable that Ser112 (Ser[112]) of Ypt1 would be targeted in a PINK1-independent manner (Fig EV5A).

Finally, we investigated whether the charged patch in Rabin8 is conserved and interacts in the same way with Rab8 homologues in yeast. Sequence alignment of Rabin8 and its yeast orthologue, Sec2, suggested that the residues forming the charged patch in Rabin8 (D203, E208, E210 and E211) are conserved in Sec2 (Fig EV5B). However, in Sec2, the patch residues follow an insertion of 14

amino acids, suggesting there may be structural differences between Sec2 and Rabin8 in this region. We therefore determined whether the charged patch in Sec2 is oriented in the same way as Rabin8 by analysing structural data on Rab8A and Sec4 with their respective exchange factors. Three PDB structures were aligned by matching chains corresponding to Sec4 (PDB code 2OCY; yeast crystal structure) and Rab8 (PDB code 4LHY; human crystal structure) to the Ypt1 chain (PDB code 2BCG; yeast crystal structure). 3D superposition strikingly revealed that whilst they are similar molecules, Rabin8 and Sec2 adopt markedly different conformations when interacting with their respective GTPases (Fig EV5C). Importantly, the curvature of Sec2's coiled coil is greater than that of Rabin8, and therefore, the distance between residue D95 in Sec2 and the putative YPT1 Ser[112] phosphosite is greater than that between the homologous residue D203 of Rabin8 and Rab8 Ser[111] (Fig EV5C). These differences suggest that the charged residues in Sec2 and Rabin8 do

not interact with their corresponding GTPases in the same way, which may be the result of coevolution in the presence of, or the lack of PINK1 and the as yet to be identified kinase. Confirmation of this, however, will involve rigorous phylogenetic analysis of the GEF superfamily, which is beyond the scope of this current study.

## Discussion

Using state-of-the-art subcellular phosphoproteomics (Trost *et al*, 2010), we have made the fundamental discovery that PINK1 upon activation by mitochondrial depolarisation regulates a family of Rab GTPases, Rab8A, 8B and 13 via phosphorylation of a highly conserved residue Ser[111]. Furthermore, biochemical and cell-based analysis of Rab8A suggests that phosphorylation at Ser[111] would impair interaction with its cognate guanine exchange factor (GEF), thereby preventing its activation (Figs 9 and 10).

Akin to other small GTPases, Rab GTPases cycle between active GTP-bound and inactive GDP-bound state that differ mainly by the conformation of two guanine nucleotide binding loops known as switch I and switch II regions (Hutagalung & Novick, 2011). In the Rab8–Rabin8 complex, there is a direct interaction of the switch II region with the GEF that is a universal feature of all currently known GTPase–GEF complexes (Guo *et al*, 2013). There are additional sites of interaction of the switch I region with Rabin8 that have also been reported for other GTPase–GEF complexes (Guo *et al*, 2013). However, there is little known about the influence of residues that lie outside these direct interfaces of interaction. Our observation that the addition of a negative charge to Ser[111] (that lies close to but distinct from the Rab8 switch II-Rabin8 interface) impairs the interaction represents a novel level of regulation of the GTPase–GEF interaction (Fig 9). In future work, it will be important to analyse preparatively phosphorylated Ser[111] Rab8 to determine the effect on Rab8–Rabin8 interaction. Furthermore, structural analysis of the Ser[111]-phosphorylated Rab8 combined with modelling studies in the presence or absence of Rabin8 may reveal the mechanism of how phosphorylation alters the Rab8–Rabin8 complex.

The region in which Ser[111] lies is known as a complementarity determining region (CDR) or also a Rab subfamily motif 3 (RabSF3) that roughly comprise the α3-β5 loop (Ostermeier & Brunger, 1999; Pereira-Leal & Seabra, 2000). In previous structures of Rab GTPases in complex with their effectors, the CDR/RabSF3 has been found to be in contact with the effector, for example the Rab3a–Rabphilin structure (Ostermeier & Brunger, 1999). CDRs may determine how some but not all effector proteins can specifically recognise and bind to one Rab sub-family in the GTP-bound state but not another (Ostermeier & Brunger, 1999). Rab GTPases are unique among small GTPases for the significant degree of complexity among their effectors (Wandinger-Ness & Zerial, 2014). For example, OCRL1 is able to interact with multiple and diverse Rab GTPases including Rab5A, Rab31, Rab35, Rab6A, Rab8A and Rab8B (Wandinger-Ness & Zerial, 2014). The CDR/RabSF3 is not required for OCRL1 binding to Rab GTPases (Hou *et al*, 2011), and consistent with this, we observed no impact of a Ser111Glu phosphomimetic of Rab8A on OCRL1$_{539-901}$ binding by gel filtration complex analysis (Fig EV3C). It may be that phosphorylation at Ser[111] within the CDR/RabSF3 may influence the interaction of Rab8A, 8B and 13 with as yet unknown effector proteins and it would be exciting in future studies

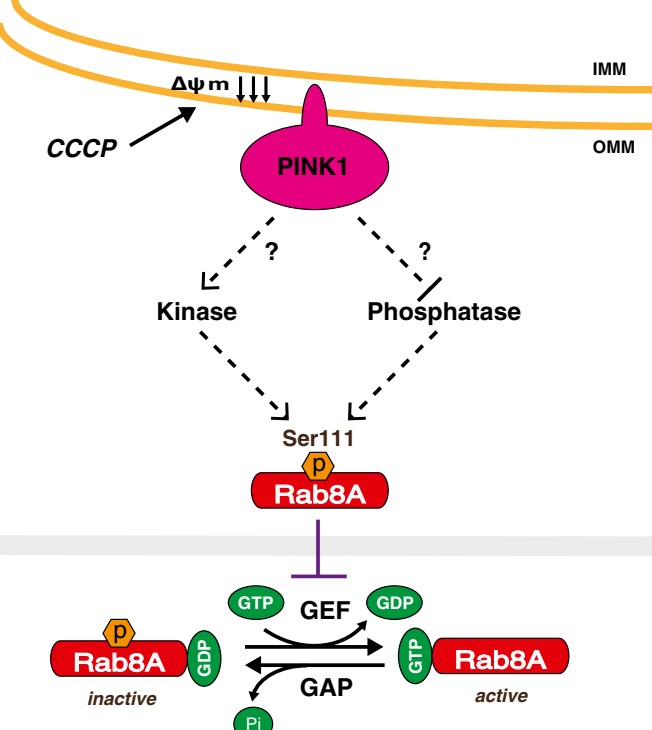

**Figure 10.  Schematic representation of PINK1 regulation of Rab8A Ser[111] phosphorylation.**
Upon mitochondrial depolarisation, PINK1 activation leads either to activation of a kinase or to inhibition of a protein phosphatase that targets Ser[111] phosphorylation of Rab8A. Phosphorylation of Rab8A at Ser[111] impairs Rabin8 (GEF)-mediated GDP exchange leading to GDP-bound inactive Rab8A.

to identify these and assess their role downstream of PINK1 activation induced by mitochondrial depolarisation.

Previously, only Rab7 together with its Rab GAPs TBC1D15 and TBC1D17 has been implicated downstream of PINK1 and has been demonstrated to be required for Parkin-mediated autophagosome formation and encapsulation of mitochondria during mitophagy (Yamano *et al*, 2014). However, our data indicate that PINK1-regulated phosphorylation of Rab GTPase Ser[111] is independent of Parkin since we observed robust phosphorylation of Rab8A in HeLa cells that lack endogenous Parkin. This suggests that PINK1 upon activation may control additional physiological processes distinct from mitophagy. Over the last few years, PINK1 has been implicated in the regulation of mitochondrial dynamics (Poole *et al*, 2008; Yang *et al*, 2008; Lutz *et al*, 2009), mitochondrial motility (Weihofen *et al*, 2009; Wang *et al*, 2011; Liu *et al*, 2012), and the generation of mitochondrial derived vesicles that selectively remove damaged mitochondrial cargos (Sugiura *et al*, 2014). Very little is known on the Rab machinery that regulates mitochondrial function and trafficking. In mammalian cells, Rab32 has been found to participate in mitochondrial dynamics and modulate mitochondria-associated membrane (MAM) properties (Alto *et al*, 2002; Bui *et al*, 2010), whilst Rab26 has been reported to mediate trafficking between mitochondria to the lysosome (Jin & Mills, 2014). Previously, Rab8A, 8B and 13 have been localised to recycling endosomes, vesicles, and early endosomes (Wandinger-Ness & Zerial, 2014), and in future

work, it will be interesting to investigate whether they are required for mitochondrial trafficking and whether this requirement is fundamentally dependent on PINK1 via Ser[111] phosphorylation.

Our data suggest that PINK1 does not directly phosphorylate Ser[111] of Rab8A, 8B and 13, suggesting that PINK1 may regulate an intermediate kinase or phosphatase that directly targets these Rab GTPases. To date, there has been very little reported on how phosphorylation of Rab GTPases alters their function. Recently, Rab5A was found to be phosphorylated by protein kinase Cε at Thr7 and this is required for Rac 1 activation, actin rearrangement, and T-cell motility (Ong *et al*, 2014). However, large-scale phosphoproteomic studies have identified multiple phosphorylation sites on Rab GTPases, suggesting that this may represent an important layer of regulation to their function. For example, Rab8A has previously been found to be phosphorylated on residues, Tyr[5], Ser[17], Thr[72], Tyr[77], Tyr[78], Ser[132], Thr[164], Ser[181], Ser[185] and Thr[192] as well as Ser[111]; however, the functional effects of phosphorylation of these sites are unknown (Olsen *et al*, 2010; Kettenbach *et al*, 2011; Shiromizu *et al*, 2013; Zhou *et al*, 2013; Bian *et al*, 2014; Sharma *et al*, 2014; Palacios-Moreno *et al*, 2015). These sites span the entire Rab GTPase protein affecting key residues required for guanine nucleotide binding, magnesium ion coordination and GTP hydrolysis as well as interaction with GEFs, GAPs and effector molecules. It will be vital to identify the upstream kinase of Ser[111] as well as for these other sites and investigate how phosphorylation modifies Rab GTPase function and whether there is any interaction between phosphorylation events on the same Rab protein. In our screen, we identified phosphopeptides for the protein kinases ICK and BRSK2 that were up-regulated 3.8- and 3.5-fold, respectively (WT/KI; H/M) (Fig EV1B, Table EV2), and in future work, it would be exciting to test whether these kinases directly target Rab GTPases at Ser[111].

Multiple sequence alignment of all 66 human Rab GTPases reveals that 15 share a serine or threonine residue at the equivalent position of Ser[111] of Rab8A within the CDR region (Appendix Fig S11). In future work, it would be interesting to determine whether the serine/threonine residue equivalent to Ser[111] of these additional Rabs can be phosphorylated and whether these are regulated by PINK1 in cells.

A major question in the field is whether PINK1-dependent pathways are linked to pathways mediated by other PD-linked genes. Pathologically, Parkinson's is defined by the presence of cytoplasmic inclusions known as Lewy bodies whose major protein component is α-synuclein. Post-mortem analysis of brains from a family with Parkinsonism harbouring PINK1 mutations has confirmed the presence of Lewy bodies in the substantia nigra (Samaranch *et al*, 2010). Furthermore, there is genetic evidence of an interaction of α-synuclein and PINK1 since mutant α-synuclein-linked pathology is exacerbated by genetic loss of PINK1 in both *C. elegans* and transgenic mouse models (Kamp *et al*, 2010; Chen *et al*, 2015). However, the molecular link between α-synuclein and PINK1 has to date remained mysterious. Previously, α-synuclein over-expression in primary rat neurons has been demonstrated to disrupt vesicular trafficking and this can be rescued by over-expression of Rab8A and related Rab GTPases (Gitler *et al*, 2008); therefore, the disruption of Rab GTPases and their downstream signalling pathway could represent a common pathway mediating PINK1 and α-synuclein-linked neurodegeneration. Furthermore, human mutations were recently

reported in the RAB39B gene in a family with an X-linked heritable Parkinsonian syndrome (Wilson *et al*, 2014) and there is now strong evidence from genome wide association studies that variation in the RAB7L locus confers significant risk to the development of sporadic PD (Nalls *et al*, 2014). Recently, LRRK2 has been implicated in the regulation of Rab GTPases. LRRK2 was found to interact with Rab7L1 to maintain retromer complex function and protein sorting (MacLeod *et al*, 2013). Mutant LRRK2 expressed in cell lines or in PD patient fibroblasts has also been reported to impair late endosomal trafficking via decreasing Rab7 activity in cells (Gomez-Suaga *et al*, 2014). Using two structurally distinct LRRK2 kinase inhibitors (Choi *et al*, 2012; Reith *et al*, 2012), we did not find any evidence that LRRK2 regulates Rab8A Ser[111] phosphorylation (Appendix Fig S12). Nevertheless, our analysis adds to an emerging picture that aberrant signalling of Rab GTPases may act as a downstream nexus for multiple genes and their corresponding pathways linked to PD-related neurodegeneration.

Our phosphoproteomic screen has also identified additional phosphopeptides that were significantly up-regulated by PINK1 including EFHD2 and FKBP38 that were increased 7.4-fold and 13.6-fold, respectively, across all four replicates (Figs 2A and EV1B and C, and Table EV2). EFHD2 is a calcium-binding adaptor protein that has been found to be associated with pathologically aggregated tau in the neurodegenerative brain in Alzheimer's disease and in a mouse model of frontotemporal dementia (Ferrer-Acosta *et al*, 2013b). To date, no study has linked EFHD2 with PD. Multiple sequence alignment of EFHD2 showed that the identified phosphosite, serine 74, is highly conserved among several species (data not shown) and lies within the N-terminal region of EFHD2 that may be important for regulating calcium-binding activity (Ferrer-Acosta *et al*, 2013a). FKBP38 is a membrane chaperone predominantly localised in mitochondria. It was recently reported that FKBP38 translocates from mitochondria to the endoplasmic reticulum during mitophagy and this escape is essential for suppression of apoptosis during mitophagy (Saita *et al*, 2013). The up-regulated phosphopeptide of FKBP38 identified in our screen lies close to the C-terminus of FKBP38 that is essential for its escape (Saita *et al*, 2013). In future work, it will be exciting to validate these phosphosites and assess how PINK1-dependent phosphorylation of these proteins alters their function and how this is linked to downstream PINK1 signalling.

Overall, our studies have identified a critical role of PINK1 in the regulation of the phosphorylation of Rab GTPases at Ser[111] and outline a novel signalling pathway for PINK1 independent of Parkin. GEFs are required for the activation of Rab GTPases, and our analysis indicates that phosphorylation would impair the activation of Rab GTPases by their cognate GEF. Our study lays the foundation for future work to uncover the identity of the upstream kinase mediating Rab Ser[111] phosphorylation as well as validating other potential targets that we have uncovered in our comprehensive analysis of PINK1-dependent proteins. Our findings should also stimulate general interest in understanding how Rab GTPases are regulated by protein phosphorylation. Our results strongly suggest that further understanding of the biological consequences of disruption of Rab GTPases will illuminate new fundamental mechanisms underlying Parkinson's. Our results also indicate that monitoring Rab8A/8B/13 Ser[111] phosphorylation represents a novel biomarker for PINK1 activity and may have clinical utility as a biomarker in PD.

# Materials and Methods

### Reagents

Tissue culture reagents were from Life Technologies. [$\gamma$-$^{32}$P] ATP was from PerkinElmer. All mutagenesis was carried out using the QuikChange site-directed mutagenesis method (Stratagene) with KOD polymerase (Novagen). All DNA constructs were verified by DNA sequencing, which was performed by The Sequencing Service, School of Life Sciences, University of Dundee, using DYEnamic ET terminator chemistry (Amersham Biosciences) on Applied Biosystems automated DNA sequencers. DNA for mammalian cell transfection was amplified in *E. coli* DH5$\alpha$ strain, and plasmid preparation was done using Qiagen Maxi prep Kit according to the manufacturer's protocol. All cDNA plasmids and antibodies generated for this study are available to request through our reagents website (https://mrcppureagents.dundee.ac.uk/). All other reagents and chemicals were standard grade from Sigma or as indicated.

### Antibodies

The following antibodies were raised by the Division of Signal Transduction Therapy (DSTT) at the University of Dundee in sheep and affinity-purified against the indicated antigens: anti-Rab8A phospho-Ser[111] (S503D, 4th bleed; raised against residues 104–117 of human Rab8A: RNIEEHApSADVEKMR); anti-Rab8B phospho-Ser[111] (S504D, 5th bleed; raised against residues 104–117 of human Rab8B: RNIEE-HApSSDVERMR); anti-Rab13 phospho-Ser[111] (S505D, 8th bleed; raised against residues 104–117 of human Rab13: KSIKENApSAG-VERLR); anti-total PINK1 (for immunoprecipitation) (S774C, 3rd bleed; raised against residues 235–end of mouse PINK1); anti-total PINK1 (for immunoprecipitation–immunoblotting) (S086D, 3rd bleed; raised against residues 175–250 of mouse PINK1), anti-total Parkin (for immunoprecipitation) (S328D, 5th bleed; raised against full-length recombinant GST-mouse Parkin); anti-total PINK1 (for immunoprecipitation) (S460C) as previously described (Kondapalli *et al*, 2012); and anti-GFP antibody (S268B, 1st bleed). The mouse monoclonal anti-PINK1 antibody (human PINK1 residues 125–539) was raised by Dundee Cell Products. Anti-Rab8A (for Rab8A specific immunoprecipitation), anti-Hsp60 and anti-GAPDH antibodies were obtained from Cell Signalling Technology. Anti-Rab8 (for immunoblotting) and anti-HA agarose bead were obtained from Sigma. GFP binder sepharose beads were generated by the DSTT. Anti-Parkin mouse monoclonal antibody was obtained from Santa Cruz. Anti-CISD1 and anti-Rabin8 (Rab3IP) antibodies were obtained from Proteintech. The rabbit monoclonal (NIAR164) anti-mitofusin 2 antibody was obtained from Abcam. Anti-HA HRP antibody was obtained from Roche. Anti-Parkin phospho-Ser[65] rabbit monoclonal antibody was raised by Epitomics in collaboration with the Michael J Fox Foundation for Research. Anti-LRRK2 and anti-LRRK2 phospho-Ser935 antibodies were obtained from Dario Alessi (Dundee).

### Cell culture

Flp-In T-Rex HEK293 cells stably expressing FLAG empty, PINK1-FLAG kinase-inactive (KI) or PINK1-FLAG wild-type (WT) were generated previously (Kondapalli *et al*, 2012). CRISPR/Cas9 system-generated PINK1 knock out (KO) HeLa cells were kindly provided by

Richard Youle (NIH). Flp-In T-Rex HEK293 cells stably expressing GFP-LRRK2 were provided by Professor Dario Alessi (University of Dundee, UK) and have been described (Dzamko *et al*, 2010). Cells were cultured in DMEM (Dulbeco's modified Eagle's medium) supplemented with 10% (v/v) foetal bovine serum, 2 mM L-glutamine, 100 U/ml penicillin and 0.1 mg/ml streptomycin at 37°C under a 5% CO$_2$ atmosphere. MEF and HeLa cells were maintained using DMEM plus 1% (v/v) non-essential amino acid. Flp-In T-Rex HEK293 cells were maintained using DMEM plus 15 $\mu$g/ml of blasticidin and 100 $\mu$g/ml of hygromycin. To express protein in Flp-In T-Rex HEK293 cells, 0.1 $\mu$g/ml of doxycycline was added to the medium for 24 h. Cell transfections were performed using polyethylenimine (Polysciences) or Lipofectamine 2000 (Life Technologies) according to the manufacturer's instruction. To uncouple mitochondria, cells were treated with 10 $\mu$M CCCP (carbonyl cyanide m-chlorophenyl-hydrazone) dissolved in DMSO for the indicated times.

### Primary human skin fibroblasts

Primary skin fibroblasts at low passage numbers (3–5) were contributed by the DNA and Cell Bank of the Institut du Cerveau et de la Moelle épinière (ICM), Hôpital de la Pitié-Salpêtrière, Paris, France. They were obtained from skin biopsies from patients with PD and age-matched healthy individuals following routine clinical procedures, underwritten informed consent and approval by a local ethics committee (Comité de Protection des Personnes "Ile de France"). Patients were screened for *PARK2* and *PINK1* mutations by exon dosage methods and bidirectional Sanger sequencing of the entire coding sequence using an ABI 3730 automated sequencer, as described previously (Periquet *et al*, 2003; Ibanez *et al*, 2006). Fibroblasts were cultured in Dulbecco's modified Eagle's medium supplemented with glucose (4.5 g/l), L-glutamine (2 mM), HEPES (10 mM), foetal bovine serum (10%) and penicillin (50 U/ml)/ streptomycin (50 $\mu$g/ml) plus 1% (v/v) non-essential amino acid and grown at 37°C in a 5% CO$_2$ atmosphere.

### Isolation and immortalisation of MEFs

Littermate matched wild-type and homozygous PINK1 or Parkin knockout mouse embryonic fibroblasts (MEFs) were isolated from mouse embryos at day E13.5 resulting from crosses between heterozygous mice using a previously described protocol (Castor *et al*, 2013). Briefly, on day E13.5, the heads were used for genotyping. The red organs were removed, and the embryo was minced and resuspended in 1 ml trypsin and incubated at 37°C for 15 min before the addition of 10 ml growth medium. Cells were plated and allowed to attach overnight before cells were washed with fresh medium to remove debris. When cells reached confluency, they were split and replated and this was considered passage 1. MEF cells were immortalised using SV40 large T antigen. All animal studies and breeding was approved by the University of Dundee ethical committee and performed under a U.K. Home Office project licence.

### Generation of Rab8A knockout cells using CRISPR/Cas9 gene editing

Analysis of the RAB8A locus (ENSG00000167461) showed a common translational start in exon 1 and potential KO CRISPR guide

pairs were subsequently identified using a sanger centre CRISPR webtool (http://www.sanger.ac.uk/htgt/wge/find_crisprs). The chosen guide pair (sense 5′-TGTTCAAGCTGCTGCTGATC and antisense 5′-ATATTACACTCTCTCCCCGA) cut as far upstream as possible to generate indels in the region containing the ATG start codon; an additional G was added to the 5′ end of each guide to maximize expression from the U6 promoter. Complementary oligos were designed and annealed to yield dsDNA inserts with compatible over-hangs to BbsI-digested vectors (Cong *et al*, 2013), the antisense guide was cloned into the spCas9 D10A expressing vector pX335 (Addgene Plasmid #42335) and the sense guide into the puromycin selectable plasmid pBABED P U6 (University of Dundee). HeLa cells were co-transfected with 1 μg of each plasmid using PEI in a 10-cm dish. Following 24 h of recovery and a further 48 h of puromycin selection (1 μg/ml) the transfection was repeated and cells subjected to a further round of puromycin selection to enrich for transfectants. The cell pool was subsequently single cell sorted by FACS and clones analysed for RAB8A depletion by immunoblotting and sequencing. Briefly, genomic DNA was isolated and the region surrounding the ATG start codon of RAB8A amplified by PCR (forward primer: TCTTCACTGC TGGTCAATCAGAGC; reverse primer: GTGGAGA TAAAAGTGGAG TTGAAGGC). The resulting PCR products were subcloned into the holding vector pSC-B (StrataClone Blunt PCR Cloning Kit, Agilent Technologies) and twelve colonies (white) picked for each clonal line. Plasmid DNAs were isolated and cut with EcoRI to verify insert size before being sent for sequencing with primers M13F and M13R. PCR products are mixed following CRISPR due to differences between the targeted alleles and we have found in prac-tice that analysis of >10 clones from a given clonal line is sufficient to verify the allelic population. Sequencing of the exon 1 PCR fragments from the knockout lines revealed a 110 base-pair deletion (including start codon) and 70 base-pair insertion (34 + 36 base-pair insertions) confirming the presence of frameshifting indels and successful KO of the RAB8A loci (data not shown).

**SILAC experiment and phosphopeptide enrichment**

Flp-In T-Rex HEK293 cells stably expressing either FLAG empty, PINK1-FLAG kinase-inactive or PINK1-FLAG wild-type were grown in "light" (K0R0), "medium" (K4R6) and "heavy" (K8R10) SILAC media, respectively, for at least 5 passages. Cells in each condition were stimulated with 10 μM CCCP for 3 h and were scraped in appropriate amount of homogenisation buffer (8.55% w/v sucrose in 3 mM imidazole pH 7.4, supplemented with protease inhibitor and phosphatase inhibitor cocktail from Roche and Benzonase from Roche). The cells were lysed by mechanical disruption using a stain-less steel homogeniser, and unbroken cells and nuclei were removed by centrifugation at 1,000 *g* for 10 min at 4°C. The membrane fraction in the remaining post-nuclear supernatant was enriched by ultra-centrifugation at 100,000 *g* for 30 min (4°C). Four biological replicates of 3 mg of these membrane fractions enriched in mitochondria were solubilised in 1% sodium 3-[(2-methyl-2-undecyl-1,3-dioxolan-4-yl)methoxy]-1-propanesulfonate (commer-cially available as RapiGest, Waters, UK), 50 mM Tris pH 8.0 and 1 mM TCEP plus phosphatase inhibitors and heated for 5 min at 70°C. After alkylation with 5 mM iodoacetamide and subsequent quenching with 10 mM DTT, solutions were diluted to 0.1% Rapi-Gest using 50 mM Tris–HCl pH 8.0 and proteins were digested by

trypsin (1:50) overnight at 37°C. Rapigest was cleaved by the addi-tion of 1% trifluoroacetic acid (TFA) and removed by solid-phase extraction. Samples were then resolubilised in 80% acetonitrile (ACN) and 0.1% formic acid and subjected to HILIC (hydrophilic interaction chromatography) (McNulty & Annan, 2008) using a TSK gel Amide-80 (4.6 mm × 25 cm) column (Tosoh, Japan). Fifteen of the later fractions, which contain the phosphopeptides, were collected in 2-min intervals and subsequently enriched for phospho-peptides using self-made $TiO_2$ spin columns (Trost *et al*, 2009).

**LC-MS/MS protein identification and quantitation**

Mass spectrometric analyses were conducted similarly as previously described (Ritorto *et al*, 2013; Dill *et al*, 2015) on an Orbitrap Velos Pro mass spectrometer coupled to an Ultimate 3000 UHPLC system with a 50 cm Acclaim PepMap 100 analytical column (75 μm ID, 3 μm C18) in conjunction with a PepMap trapping column (100 μm × 2 cm, 5 μm C18) (all Thermo-Fisher Scientific). Acquisition settings were as follows: lockmass of 445.120024, MS1 with 60,000 resolution, top 20 CID MS/MS using Rapid Scan, monoisotopic precursor selection, unassigned charge states and $z = 1$ rejected, and dynamic exclusion of 60 s with repeat count 1. Normalised collision energy was set to 35, and activation time was 10 ms. Four-hour linear gradients were performed from 5% solvent B to 35% solvent B (solvent A: 0.1% formic acid, solvent B: 80% acetonitrile 0.08% formic acid) at 300 nl/min in 217 min with a 23-min washing and re-equilibration step.

Protein identification and quantification were made using MaxQuant (Cox & Mann, 2008) version 1.3.0.5 with the following parameters: FT mass tolerance 20 ppm; MS/MS ion trap tolerance 0.5 Da; trypsin/P set as enzyme; stable modification carbamido-methyl (C); variable modifications, oxidation (M), acetyl (protein N-term) and phospho (STY); maximum 5 modifications per peptide; and 2 missed cleavages. Searches were conducted using a combined UniProt-Trembl *Homo sapiens* database with isoforms downloaded on 15 February 2012 plus common contaminants (117,706 sequences). Identifications were filtered at a 1% FDR at the peptide level, accepting a minimum peptide length of 7. Quantification required a minimum ratio count of 2. Requantification was enabled, and match between runs was allowed within a 5-min window. Normalised ratios for peptides showed a median variability of 25–28% (ratio variation for 95% of the ratios was below 75%). Downstream analyses were performed in Perseus 1.4.0.20 (Cox & Mann, 2012) where statistical tests (one-sample *t*-test, $P < 0.05$) for each ratio (H/L, H/M, M/L) were performed. The mass spectrometry raw data and the Maxquant output from this publication have been submitted to the PRIDE database (Vizcaino *et al*, 2013) (https://www.ebi.ac.uk/pride/archive/) and assigned the identifier PXD002127.

**LC-MS/MS mapping of in-gel tryptic digested Rab8A, 8B and 13 Ser[111] phosphopeptides**

Samples were analysed on a linear ion trap–orbitrap hybrid mass spectrometer (Orbitrap-Classic, Thermo) equipped with a nano-electrospray ion source (Thermo) and coupled to a Proxeon EASY-nLC system. Peptides were injected onto a Thermo (Part No. 160321) Acclaim PepMap100 reversed-phase C18 3 μm column, 75 μm × 15 cm, with a flow of 300 nl/min, and eluted with a

45-min linear gradient of 95% solvent A (2% acetonitrile, 0.1% formic acid in H$_2$O) to 40% solvent B (90% acetonitrile, 0.08% formic acid in H$_2$O), followed by a rise to 80% solvent B at 48 min. The instrument was operated with the "lock mass" option to improve the mass accuracy of precursor ions, and data were acquired in the data-dependent mode, automatically switching between MS and MS-MS acquisition. Full-scan spectra (m/z 340–2,000) were acquired in the orbitrap with resolution $R = 60,000$ at m/z 400 (after accumulation to an FTMS Full AGC Target; 1,000,000; MSn AGC Target; 100,000). The 5 most intense ions, above a specified minimum signal threshold (5,000), based upon a low resolution ($R = 15,000$) preview of the survey scan, were fragmented by collision-induced dissociation and recorded in the linear ion trap (Full AGC Target: 30,000; MSn AGC Target: 5,000). Multi-stage activation was used to provide a pseudo MS3 scan of any parent ions showing a neutral loss of 48.9885, 32.6570 and 24.4942, allowing for 2+ , 3+ and 4+ ions respectively. The resulting pseudo MS3 scan was automatically combined with the relevant MS2 scan prior to data analysis. Extracted ion chromatograms (XICs) were obtained using Xcalibur software (Thermo). Automatic processing was employed with parameters as follows: Gaussian smoothing, 7 points; no baseline subtraction; mass tolerance $\pm$ 10.0 ppm; and mass precision, 4 decimals.

## Immunoblotting and immunoprecipitation

Protein lysates were extracted in lysis buffer containing buffers 50 mM Tris–HCl (pH 7.5), 1 mM EDTA, 1 mM EGTA, 1% (w/v) Triton, 1 mM sodium orthovanadate, 10 mM sodium glycerophosphate, 50 mM sodium fluoride, 10 mM sodium pyrophosphate, 0.25 M sucrose, 0.1% (v/v) 2-mercaptoethanol, 1 mM benzamidine, 0.1 mM PMSF and protease inhibitor cocktail (Roche). Lysates were clarified by centrifugation at 17,000 $g$ for 15 min at 4°C, and the supernatant was collected. Protein concentration was determined using the Bradford method (Thermo Scientific) with BSA as the standard.

For immunoprecipitation of HA-tagged Rab proteins, 0.25–1 mg of protein extracts was undertaken by standard methods with anti-HA agarose beads. For immunoprecipitation of endogenous Rab8A, cell lysates containing 1 mg of protein were immunoprecipitated at 4°C for at least 2 h with 2 μl of anti-Rab8A antibody pre-bound to 15 μl of protein A agarose beads. The immunoprecipitates were washed three times with lysis buffer containing 0.15 M NaCl and eluted by resuspending in 20 μl of 1× SDS sample buffer.

Immunoprecipitates or cell extracts (25–50 μg of protein) were subjected to SDS–PAGE (4–12%) and transferred on to nitrocellulose membranes. Membranes were blocked for 1 h in Tris-buffered saline with 0.1% Tween (TBST) containing 5% (w/v) BSA. Membranes were probed with the indicated antibodies in TBST containing 5% (w/v) BSA overnight at 4°C. Detection was performed using appropriate HRP-conjugated secondary antibodies and enhanced chemiluminescence reagent.

## Mitochondrial protein enrichment and ubiquitylated mitochondrial protein capture

Mitochondrial proteins were enriched as described previously (Kazlauskaite *et al*, 2015). For ubiquitylated protein capture, 200 μg

of mitochondrial protein extracts was used for pull down with HALO-UBA$^{UBQLN1}$ resin as described previously (Kazlauskaite *et al*, 2015).

## Structural and bioinformatics analysis

### Phosphorylation sites

Full-length protein sequences for ubiquitin, Parkin, Rab1A and Rab8A were retrieved from UniProt via the EMBL-EBI database retrieval service (Lopez *et al*, 2003), client with the Jalview Desktop (Waterhouse *et al*, 2009) and manually aligned to match observed phosphorylation site positions. Representative structures for phosphorylated regions were identified via UniProt annotation and the PDBe SIFTS service (Velankar *et al*, 2013), and structures for ubiquitin (PDB code: 2W9N chain A), Parkin (PDB code: 1IYF model 1 chain A), Rab1A (PDB code: 3TKL chain A) and Rab8A (PDB code: 4LXH chain A) were downloaded. Structures were visualised in UCSF Chimera (Pettersen *et al*, 2004) and were superimposed with UCSF Chimera's match command using the C-α positions for the observed site and adjacent two amino acids on either side. Detailed descriptions of local secondary structure conformations at these locations were obtained from PDBsum (de Beer *et al*, 2014), and the visualisation was rendered with POVray (bundled with UCSF Chimera).

### Human and Yeast Rab8 GTPases and GEFs

Multiple sequence alignment of yeast Ypt1, yeast Sec4 and human Rab8A was made with T-COFFEE (default settings, v8.99). Alignment of the yeast Sec2 and human Rabin8 sequences was performed with Jalview's pairwise alignment function. Alignment of PDB structures containing Ypt1, Sec4 and Rab8A was made with UCSF Chimera's matchmaker function (v 1.10.1 with default parameters).

## Kinase assays and phosphorylation site mapping

For *in vitro* kinase assay, 1.2 μg of recombinant WT or S111A mutant Rab8A, or 0.5 μg of ubiquitin as a positive control was incubated with 1.1 μg of *E. coli*-expressed WT or KI (D359A) MBP-fused TcPINK1 in total 10 μl of kinase buffer containing 50 mM Tris/HCl (pH 7.5), 0.1 mM EGTA, 10 mM MgCl$_2$, 2 mM DTT and 0.1 mM [γ-$^{32}$P] ATP (approx. 500 cpm/pmol) at 30°C with continuous shaking. Reactions were terminated by adding SDS sample buffer at the time indicated. The reaction mixtures were then resolved by SDS–PAGE. Proteins were detected by colloidal Coomassie blue staining and dried completely using a gel dryer (Bio-Rad Laboratories). Incorporation of [γ-$^{32}$P] into substrates was analysed by autoradiography using Amersham hyper-sensitive film. Cerenkov counting was used to calculate the stoichiometry of substrate phosphorylation as mol of [γ-$^{32}$P] incorporation/mol of substrate.

For mapping the site on Rab8A phosphorylated by TcPINK1, recombinant Rab8A (24 μg) was incubated with MBP-fused WT TcPINK1 (50 μg) for 120 min in the same condition as the kinase assay, except [γ-$^{32}$P] ATP that was approx. 20,000 cpm/pmol. The reaction was terminated by the addition of SDS sample buffer with 10 mM DTT, boiled and subsequently alkylated with 50 mM iodoacetamide before samples were subjected to electrophoresis on a Bis-Tris 4–12% polyacrylamide gel, which was then stained with colloidal Coomassie blue (Invitrogen). Protein bands were excised

from the gel, and 98% of the $^{32}$P radioactivity incorporated into Rab8A was recovered from the gel bands after tryptic digestion. Peptides were chromatographed on a reversed-phase HPLC Vydac C18 column (catalogue number 218TP5215, Separations Group) equilibrated in 0.1% trifluoroacetic acid, and the column developed with a linear acetonitrile gradient at a flow rate of 0.2 ml/min before fractions (0.1 ml each) was collected and analysed for $^{32}$P radioactivity by Cerenkov counting. Isolated phosphopeptides were analysed by LC–MS/MS on a Thermo U3000 RSLC nano-LC system coupled to a Thermo LTQ-Orbitrap Velos Pro mass spectrometer. The resultant data files were searched using Mascot (www.matrixscience.com) run on an in-house system against a database containing the Rab8A sequence, with a 10 ppm mass accuracy for precursor ions and a 0.6 Da tolerance for fragment ions and allowing for phospho (S/T), phospho (Y), oxidation (M) and dioxidation (M) as variable modifications. Individual MS/MS spectra were inspected using Xcalibur v2.2 software (Thermo Scientific). The site of phosphorylation of these $^{32}$P-labelled peptides was determined by solid-phase Edman degradation on a Shimadzu PPSQ33A sequencer of the peptide coupled to Sequelon-AA membrane (Applied Biosystems).

### Protein expression and purification

The wild-type (WT) or kinase-inactive *Tribolium castaneum* PINK1 (TcPINK1) was expressed in *E. coli* and purified as described previously (Woodroof *et al*, 2011). The Rab8A WT was expressed in *E. coli* BL21(DE3) and purified as described previously (Bleimling *et al*, 2009). The S111E and S111A substitutions of Rab8A were introduced by site-directed mutagenesis (QuikChange, Agilent Technologies, Santa Clara, CA, USA), and proteins were expressed and purified analogously to Rab8A WT. The expression and purification of the Rabin8$_{153–237}$ and OCRL1$_{539–901}$ were performed as described in the study by Guo *et al* (2013) and Hou *et al* (2011). His-halo-ubiquilin1 UBA domain tetramer (UBA$^{UBQLN1}$) was expressed in *E. coli* BL21 cells and purified as described previously (Kazlauskaite *et al*, 2015).

### Analytical size-exclusion chromatography

OCRL1$_{539–901}$ (15 μM) and individual Rab proteins (19.5 μM) were incubated for 1 h in a volume of 70 μl and subjected to chromatographic separation on a Superdex 200 (10/30) gel filtration column (GE Healthcare, USA) using a HPLC system (Shimadzu, Japan) equipped with a SPD-20AV UV/Vis detector and detected at 254 nm. The column was pre-equilibrated with 20 mM HEPES pH 7.5, 50 mM NaCl, 2 mM DTE, 1 mM MgCl$_2$ and 1 μM GppNHp.

### Rabin8 catalysed nucleotide exchange assay

The Rab8A-GDP variants were loaded with the fluorescent GDP analogue 2′/3′-(*N*-methylanthraniloyl)-GDP (mantGDP). The loading was performed with 5-fold excess over the protein of mantGDP in the presence of 5 mM EDTA for 2 h at room temperature in the dark. Rabin8-catalysed mantGDP release was measured at 25°C with a Fluoromax-4 fluorescence spectrometer (HORIBA Jobin Yvon), excited at $\lambda_{exc} = 365$ nm and monitored at $\lambda_{em} = 440$ nm. The Rab proteins (1 μM) were incubated with 100 μM GDP in 1 ml buffer (20 mM HEPES pH 7.5, 50 mM NaCl, 1 mM MgCl$_2$, 2 mM DTE) in a Quartz SUPRASIL cuvette (Hellma Analytics, Germany), and the reaction was started by the addition of 0.5 μM Rabin8. The decrease in mant fluorescence was used as a measure of mantGDP release.

### Thermal shift assay

The thermal shift assay can be used to investigate the stability of proteins (Ericsson *et al*, 2006). The melting point of the protein is determined by the fluorescence of a dye (Sypro Orange, Sigma-Aldrich, USA). The fluorescence of the dye is quenched in solution but remains when the dye is bound to hydrophobic regions. Through a successive increase in temperature, the protein unfolds and exposes more hydrophobic regions that the dye can bind to. This leads to an increase in fluorescence. The assay was performed with 1 and 10 μg of the Rab proteins, respectively. The proteins were mixed in a 1:1 ration with a 10× Sypro Orange solution in a total volume of 20 μl. The probes were prepared as triplicates, and the assay was performed in a 96-well plate in a RT–PCR cycler (Agilent Technologies Stratagene Mx3000P).

### Intrinsic GTP hydrolysis

The Rab proteins (1 mg) were loaded with GTP by incubation with a 20-fold excess of GTP and 5 mM EDTA for 2 h at RT. After the loading, excess GTP was removed by applying the protein solution to a Nap10 column (GE Healthcare, USA) and subsequent washing with buffer (20 mM HEPES pH 7.5, 50 mM NaCl, 1 μM GTP, 2 mM DTE) according to the manufacturer's manual. The analysis of intrinsic GTP hydrolysis was performed with 50 μM protein. At defined time points, 20 μl of the protein solution was denatured by incubation for 10 min at 95°C and subsequently centrifuged to separate the protein and the nucleotide. An isocratic elution (50 mM potassium phosphate, pH 6.6, 10 mM tetra-*N*-butylammonium bromide, 12% (v/v) acetonitrile) was used to separate GDP and GTP on a Prontosil C18 120-5-C18-AQ column (Bischoff chromatography). The peak areas of GTP and GDP were used as a measure of GTP hydrolysis.

### Rab-GAP assay

Rab8 proteins (1 mg) were loaded with GTP by incubation with a 20-fold excess of GTP and 5 mM EDTA for 2 h at RT. After the loading, excess GTP was removed by applying the protein solution to Nap10 columns (GE Healthcare, USA) and subsequent washing with buffer (20 mM HEPES pH 7.5, 50 mM NaCl, 1 μM GTP, 2 mM DTE) according to the manufacturer's manual. The analysis of the GAP-stimulated GTP hydrolysis was performed with 30 μM of the respective Rab8 variant and 100 nM of TBC1D20. At defined time points, 20 μl of the protein solution was denatured by incubation for 10 min at 95°C and subsequently centrifuged to separate the protein and the nucleotide. An isocratic elution (50 mM potassium phosphate, pH 6.6, 10 mM tetra-*N*-butylammonium bromide, 12% (v/v) acetonitrile) was used to separate GDP and GTP on a Prontosil C18 120-5-C18-AQ column (Bischoff chromatography). The peak areas of GTP and GDP were used as a measure of GTP hydrolysis.

**Expanded View** for this article is available online:
http://emboj.embopress.org

## Acknowledgements

We thank Ian Ganley (Dundee) for helpful discussions, Richard Youle (NIH) for HeLa PINK1 knockout cell lines and Dario Alessi (Dundee) for GFP-LRRK2 Flp-In T-Rex HEK293 cells and LRRK2 antibodies. We also thank Maria Rosenegger for invaluable technical support. We thank Sylvie Forlani (Paris) for fibroblast collection and banking. We thank Miguel Martins (Leicester) for sending cryopreserved embryos of PINK1 knockout mice. We thank Thomas McWilliams and Agne Kazlauskaite for generating the PINK1 and Parkin MEFs. We are grateful to the sequencing service (College of Life Sciences, University of Dundee), and James Hastie and Hilary McLauchlan and the antibody purification and protein production teams (Division of Signal Transduction Therapy (DSTT), University of Dundee) for excellent technical support. UCSF Chimera is developed by the Resource for Biocomputing, Visualization, and Informatics at the University of California, San Francisco (supported by NIGMS P41-GM103311). M.M.K.M. is funded by a Wellcome Trust Senior Research Fellowship in Clinical Science (101022/Z/13/Z). M.T. is funded by the Medical Research Council (MRC), UK (MC_UU_12016/5). J.B.P. is supported by the BBSRC BBR Grant (BB/L020742/1). A.I. and R.L. acknowledge funding from the German Research Foundation (DFG: SFB1035, project B05). O.C. acknowledges funding from Investissements d'avenir—ANR-10-IAIHU-06. This work was supported by the Medical Research Council; the Wellcome Trust; Parkinson's UK; the Michael J. Fox Foundation for Parkinson's disease research; Tenovus Scotland; and a Wellcome/MRC PD consortium grant to UCL Institute of Neurology, University of Sheffield and MRC-PPU of University of Dundee. We also thank the pharmaceutical companies supporting the Division of Signal Transduction Therapy Unit (AstraZeneca, Boehringer-Ingelheim, GlaxoSmithKline, Merck KGaA, Janssen Pharmaceutica and Pfizer) for financial support.

## Author contributions

YCL and CK performed most of the experiments. RL performed biochemical analysis of Rab8A GTPase function under supervision of AI. JBP performed bioinformatic analysis. BDD, RG and DGC assisted with mass spectrometry analysis under supervision of MT. HIW performed biochemical analysis of PINK1. MP generated cDNA constructs used in the project. TJM designed and generated Cas9/CRISPR oligos. OC and JCC generated and provided PINK1 patient fibroblasts. YCL, CK, AI, MT and MMKM planned experiments and analysed results. YCL and MMKM wrote the paper with contribution from all the authors. MT and MMKM conceived and supervised the project.

## Conflict of interest

The authors declare that they have no conflict of interest.

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
