## [Review Process File · The EMBO Journal]

Manuscript EMBO-2015-91593

Phosphoproteomic screening identifies Rab GTPases as novel downstream targets of PINK1

Yu-Chiang Lai, Chandana Kondapalli, Ronny Lehneck, James B Procter, Brian D Dill, Helen I Woodroof, Robert Gourlay, Mark Peggie, Thomas J Macartney, Olga Corti, Jean-Christophe Corvol, David G Campbell, Aymelt Itzen, Matthias Trost and Miratul MK Muqit

Corresponding author: Miratul Muqit, University Dundee

Review timeline:

Submission date:	29 April 2015
Editorial Decision:	21 May 2015
Revision received:	25 August 2015
Editorial Decision:	07 September 2015
Revision received:	14 September 2015
Accepted:	18 September 2015

Editor: Hartmut Vodermaier

Transaction Report:

1st Editorial Decision

21 May 2015

Thank you again for submitting your manuscript on phosphoproteomic analysis of PINK1 targets for our consideration. We have now received comments from three expert reviewers, copied below for your information. As you will see, the referees appreciate the timeliness of this research as well as the overall technical quality of your analyses, but they also retain some reservations regarding the depth of overall insight provided by your results.

While we feel that elucidating the exact role of Rab GTPase phosphorylation downstream of PINK1, as requested by referee 1, would probably exceed the scope of this primary study, especially when considering it as a resource article, referees 2 and 3 however raise a number of well-taken more specific points that in our opinion would greatly improve the decisiveness and insightfulness of the present manuscript. In this respect, especially the last point of referee 3 (regarding LRKK2) would appear particularly important to follow up on with some additional experimentation. Similarly, referee 2's points 3 (regarding effects on GAP-stimulated hydrolysis) and 4 (regarding effects on Rab GTPase cellular activation, e.g. by proxy of localization of fluorescently tagged wild-type and mutant Rab8) should be experimentally addressed. On the other hand, while any data in response to referee 2's point 5 (concerning Rab GTPase contribution to PINK1 loss-of-function phenotypes) would certainly be very valuable, I understand that obtaining decisive phenocopy data may be difficult to achieve in the absence of dedicated new animal models and could not be expected during a regular revision.

Should you be able to address the above-mentioned key points, as well as to adequately answer the

remaining more specific technical and/or discussion points raised by referees 2 and 3, then we should be happy to consider a revised manuscript further for publication in The EMBO Journal. Since it is our policy to allow only a single round of major revision, please note that it will however be important to carefully respond to all points raised during this round. We generally grant three months as standard revision time, and our 'scooping protection' policy means that any competing manuscripts published during this period will have no negative impact on our final assessment of your revised study; but please do contact us in case you should become aware of upcoming competing work, or if you should have difficulties meeting the three-month deadline, in order to discuss how to best proceed further. Additional information on preparing and uploading your revision can be found below.

Thank you again for the opportunity to consider this work for The EMBO Journal, and please do not hesitate to contact me should you have any comments or questions regarding the referee reports or this decision. I look forward to your revision.

REFEREE REPORTS

Referee #1:

The authors nicely show Rab8 phosphorylation downstream of PINK1 activation following mitochondrial uncoupling. The rigorous assessment is refreshing considering some of the prior literature on PINK1 substrates. However, the kinase that phosphorylates Rab8 remains unknown and how PINK1 interfaces with a downstream kinase or phosphatase is a mystery. Most problematic, the role Rab8 phosphorylation plays in mitophagy or any other activity of PINK1 or any link to Parkinson's disease is not explored. Thus, although the well-validated new downstream target of PINK1 appears to be an important clue to an unknown activity of PINK1, without some indication of what Rab GTPases are doing in relation to PINK1 activity, the work remains too preliminary for EMBO J. A better understanding of the role of Rab activity in PINK1 function is necessary.

Referee #2:

This work establishes that PINK activation leads indirectly to the phosphorylation of Rabs 8 and 13 and that this inhibits the interaction of Rab8 with its GEF Rabin8. The study is technically well done and I have no criticism of the work that is presented. There are nevertheless several areas that must be explored in greater depth so that the physiological significance of the work can be evaluated:

1. The authors show that it is unlikely that Rab8 is phosphorylated directly by PINK and they argue that PINK probably acts by phosphorylating another kinase or phosphatase to control Rab phosphorylation. Are there any candidates for kinases or phosphatases that are differentially phosphorylated upon PINK activation? They do not explicitly state whether this is or is not the case regarding the results of their phosphoproteomic analysis. If it is not the case, some speculation regarding why candidates were not picked up should be added to the Discussion.
2. More thorough evolutionary analysis of this regulatory pathway could be informative. It has been shown that expression of alpha-synuclein in yeast is toxic and that this can be suppressed by Ypt1 overexpression. Does yeast have PINK1? If so, does Ypt1 (or Sec4) have a ser at the same site? Is the charged patch on Rabin8 that purportedly interferes with binding to phosphomimetic-Rab8 conserved in Sec2? If not, is there a correlation through evolution with the presence of PINK1 and the presence of the Rab S111 and the presence of the charged patch on Rabin8?
3. The authors show that phosphomimetic-Rab8 hydrolyzes GTP at the same rate as WT. What about GAP-stimulated GTP hydrolysis? The intrinsic hydrolysis rate is not as relevant *in vivo* as the GAP-stimulated rate.
4. The effects of Rab8 phosphorylation are only studied *in vitro* with pure components. The authors need to demonstrate the effects of PINK1 activation and the S111E mutation on Rab8 function *in vivo*. Otherwise the physiological significance of this entire study is in question. Is Rab8S111E mislocalized, is WT Rab8 mislocalized upon PINK activation? A failure to activate Rab8, as proposed, should lead to its mislocalization. Effects should also be observed on Rab8-regulated membrane trafficking.
5. In the introduction the authors argue for additional PINK1 targets by pointing out that the PINK1

KO phenotype differs from that of the Parkin KO. Can the phenotype of the PINK1 null be phenocopied with a Parkin KO plus a Rab8 S111A mutation?

Referee #3:

The manuscript from Lai and colleagues describes the systematic investigation of phosphorylation targets downstream of PINK1. Using SILAC mass spectrometry (MS), the authors confirmed the known PINK1 target (ubiquitin) while uncovering a series of putative substrates in the Rab family who are phosphorylated on a single conserved Ser residue. As follow up, the authors confirmed pSer111 for each of Rab8A, Rab8B and Rab13 in a single protein gel-based MS assay captured using an HA-epitope tagged substrates. Further confirmation was obtained through generation of a pSer111 specific antibody. The series of biochemical studies using the phosphospecific Ab have been systematically carried out in WT and PINK1 $-/-$ cells rescued with either WT of inactive PINK1 protein in the presence and absence of the CCCP activating stimulus. In addition to demonstrating that exogenous PINK can trigger phosphorylation of exogenous/endogenous Rab proteins, they subsequently present timecourse analysis confirming that endogenous PINK1 can likewise trigger phosphorylation of Rab proteins. This timecourse study was particularly revealing because it showed, contrary to initial expectations, that PINK1 likely does not phosphorylate Rab proteins directly and instead that a secondary kinase must be activated. The manuscript text reads clearly and the experiments are effectively presented. The missing link is of course the PINK1 activated kinase that directly targets the Rab8a/8b/13/1A Ser111. Notwithstanding this finding, the work will be of interest to the neurodegeneration and mitochondrial biology fields and warrants consideration. Detailed comments and questions are below:

- Have the authors submitted the raw MS data from global phosphorylation and single protein analyses to a publically available repository?
- Data in Suppl Figure 1 seems to suggest that an unexpectedly high fraction of proteins remains only 80-90% labeled in the SILAC analysis even after 5 passages. If the cells have gone through >5 doublings, the unlabeled fraction of a would be expected to be $< 3\%$ just based on dilution. Is there a particular reason for this? Analytical error, reagent impurity, etc?
- The initial phosphopeptide screen revealed several additional top hits including DLST, KBTBD11 and FKBP38 protein. It would be valuable if the authors to provide some comments on these in the discussion.
- In Fig. 3, the top panels from the MS assay show the extracted precursor ion signal for the pSer111 peptides. It would be useful for the reader if this figure or a corresponding Suppl Figure would present the comparable unphosphorylated extracted ion chromatograms of this and/or another unmodified peptide for comparison. What mass tolerance was used for generating these MS1 ion signals? What did the Coomassie gel look like and what gel region was excised for the analysis. Addition of these additional elements in a parallel Suppl figure would be informative. Moreover, while a selection of the methods for this were currently presented in the legend, a more comprehensive description should be provided within the Methods section.
- The authors are likely aware that LRRK2 is a Parkinson's Disease gene, kinase and RAB GTPase containing protein. Interestingly, this protein has been shown to undergo autophosphorylation on Ser1443, the residue equivalent to Ser111 in Rab8a/8b/13/1a. Is it possible that LRRK2 serves as the kinase downstream of PINK1 in the model systems used within this paper? Can exogenously expressed LRRK2 stimulate phosphorylation of Rab8a/8b/13/1a on Ser111 in the assay systems used in Figs 3-4?

1st Revision - authors' response

25 August 2015

Major Points raised by Reviewer 1

The authors nicely show Rab8 phosphorylation downstream of PINK1 activation following mitochondrial uncoupling. The rigorous assessment is refreshing considering some of the prior

literature on PINK1 substrates. However, the kinase that phosphorylates Rab8 remains unknown and how PINK1 interfaces with a downstream kinase or phosphatase is a mystery. Most problematic, the role Rab8 phosphorylation plays in mitophagy or any other activity of PINK1 or any link to Parkinson's disease is not explored. Thus, although the well-validated new downstream target of PINK1 appears to be an important clue to an unknown activity of PINK1, without some indication of what Rab GTPases are doing in relation to PINK1 activity, the work remains too preliminary for EMBO J. A better understanding of the role of Rab activity in PINK1 function is necessary.

We agree with Reviewer 1 that it would of interest to identify the upstream kinase, however, we believe that this is beyond the scope of this initial discovery paper. We also agree that it would be important to explore the role of Rab8A phosphorylation in downstream PINK1 signalling. To address this we have generated genetic knockouts of Rab8A in cell lines using CRISPR/Cas9 technology and monitored Parkin substrate ubiquitylation at the mitochondria in response to CCCP-induced mitochondrial depolarisation that activates PINK1 kinase activity. Our analysis indicates that Rab8A is dispensable for PINK1-induced activation of Parkin E3 ligase activity. This data is now included in Fig 6.

We agree that it would be important to explore the link of Rab8A phosphorylation in Parkinson's disease. In that regard we have investigated the role of Rab8A phosphorylation in human Parkinson's disease patient fibroblasts bearing PINK1 mutations and our analysis demonstrates that Rab8A phosphorylation is totally disrupted in the PINK1 mutant fibroblasts upon treatment with CCCP. Furthermore, we have assessed Rab8A phosphorylation in a PINK1 knockout mouse model and consistent with the human genetic data, we observe disruption of Rab8A phosphorylation in PINK1 knockout mouse embryonic fibroblasts compared to wild-type controls. These data have now been included in Figure 5.

Major Points raised by Reviewer 2

1. The authors show that it is unlikely that Rab8 is phosphorylated directly by PINK and they argue that PINK probably acts by phosphorylating another kinase or phosphatase to control Rab phosphorylation. Are there any candidates for kinases or phosphatases that are differentially phosphorylated upon PINK activation? They do not explicitly state whether this is or is not the case regarding the results of their phosphoproteomic analysis. If it is not the case, some speculation regarding why candidates were not picked up should be added to the Discussion.

We have identified phosphopeptides in two protein kinases ICK and BRSK2 that were up-regulated in our screen upon PINK1 activation. We have discussed this in a paragraph on page 15 of the Discussion but believe that their validation and characterization as Rab8A kinases is beyond the scope of the current paper.

2. More thorough evolutionary analysis of this regulatory pathway could be informative. (1) It has been shown that expression of alpha-synuclein in yeast is toxic and that this can be suppressed by Ypt1 overexpression. (2) Does yeast have PINK1? (3) If so, does Ypt1 (or Sec4) have a ser at the same site? Is the charged patch on Rabin8 that purportedly interferes with binding to phosphomimetic-Rab8 conserved in Sec2? (5) If not, is there a correlation through evolution with the presence of PINK1 and the presence of the Rab S111 and the presence of the charged patch on Rabin8?

(1) Since yeast does not have endogenous alpha-synuclein, we believe this is not a physiologically relevant study.

(2 - 4) There are no true orthologs of PINK1 in yeast. We verified this by first examining the entry for PINK1 in the EggNOG[1] ortholog database, which suggests PINK1 is only found in metazoans. We also employed the EggNOG hidden markov model for PINK1 to search the NCBI NR protein sequence database with the EMBL-EBI HMMER3 server[2], which yielded no significant matches in Saccharomyces.

(5). The negative surface patch of Rabin8 adjacent to the Rab8A interaction interface is comprised of residues Asp187 (D187), Glu192 (E192), Glu194 (E194) and Glu 195 (E195) (Guo et al, 2013). Given the functional relationship between Rab8A Ser¹¹¹ phosphorylation and the Rabin8 negative patch, we have undertaken bioinformatic analysis to determine whether this interaction has co-evolved with PINK1. Were that the case, then for orthologues of Rab8 and Rabin8 in organisms that lack PINK1, the interaction between the charged patch and Ser¹¹¹ would not need to be conserved. To explore this hypothesis, we examined proteins orthologous to Rab8 and Rabin8 in yeast.

Our analysis reveals structural differences that suggest that the charged residues in Sec2 and Rabin8 do not interact with their corresponding GTPases in the same way; which may be the result of coevolution in the presence of, or lack of PINK1 and the as yet to be identified kinase. Confirmation of this, however, will involve rigorous phylogenetic analysis of the GEF superfamily, which is beyond the scope of this current study.

This new bioinformatic analysis is reported in the Results section on page 12 and 13 and the data shown in Figure EV5.

3. The authors show that phosphomimetic-Rab8 hydrolyzes GTP at the same rate as WT. What about GAP-stimulated GTP hydrolysis? The intrinsic hydrolysis rate is not as relevant in vivo as the GAP-stimulated rate.

We have now tested the effect of the Rab8 phosphomimetic on GAP-stimulated GTP hydrolysis and do not observe any difference from the wild-type Rab8 protein. This data is now included in Figure EV3D.

4. The effects of Rab8 phosphorylation are only studied in vitro with pure components. The authors need to demonstrate the effects of PINK1 activation and the S111E mutation on Rab8 function in vivo. Otherwise the physiological significance of this entire study is in question. Is Rab8S111E mislocalized, is WT Rab8 mislocalized upon PINK activation? A failure to activate Rab8, as proposed, should lead to its mislocalization. Effects should also be observed on Rab8-regulated membrane trafficking.

We have undertaken cell-based analysis to investigate the effect of Rab Ser111 phosphorylation on Rabin8A interaction. By co-immunoprecipitation analysis we confirm that wild-type Rab8A and Rabin8 interact however, we observed that the Rabin8 interaction is significantly impaired with the phosphomimetic S111E Rab8A mutant. This data has been included in Figure 9C and Figure EV4.

With regards to other analyses of the effect of Ser111 phosphorylation in cells, the Ser residue lies in a region predicted to be involved in effector binding as well as Rabin8 interaction. Therefore we believe that to rigorously study the role of phosphorylation of Rab8A in membrane trafficking would require knowledge of the key effectors of Rab8A in the context of mitochondrial depolarisation which is currently unknown and which we believe is beyond the scope of this current paper. Future work will be directed at uncovering this.

5. In the introduction the authors argue for additional PINK1 targets by pointing out that the PINK1 KO phenotype differs from that of the Parkin KO. Can the phenotype of the PINK1 null be phenocopied with a Parkin KO plus a Rab8 S111A mutation?

That is a very interesting question. In future work it would be exciting to generate a RabSer111 knock-in mouse to cross with the parkin knockout however, believe that this is beyond the scope of the current study.

Major Points raised by Reviewer 3

- Have the authors submitted the raw MS data from global phosphorylation and single protein analyses to a publically available repository?

Yes – the data are available via ProteomeXchange with identifier PXD002127. We have stated this in the methods section.

- Data in Suppl Figure 1 seems to suggest that an unexpectedly high fraction of proteins remains only 80-90% labeled in the SILAC analysis even after 5 passages. If the cells have gone through >5 doublings, the unlabeled fraction of a would be expected to be < 3% just based on dilution. Is there a particular reason for this? Analytical error, reagent impurity, etc?

We agree with the reviewer that the figure is indeed confusing. We were using a script from the Mann lab which appears to show lower isotope incorporation rates – however, these proteins are actually just of low abundance, thus, the signal of the peptide peaks compared to noise level is <10. As such they appear to have lower isotope incorporation. This is something we have always observed in our SILAC projects – even with cells that were grown for months in SILAC medium. The critical part of this figure is the peak-top of the distribution which gives a good indication of the average incorporation which is in all cases >96%. As the graphs show also for some proteins >100% isotope incorporation (which also stems from a similar error), we have decided to remove this supplementary figure in order to not confuse readers.

- The initial phosphopeptide screen revealed several additional top hits including DLST, KBTBD11 and FKBP38 protein. It would be valuable if the authors to provide some comments on these in the discussion.

We have now added a paragraph to our Discussion on page 16 that discusses additional hits including EFHD2 and FKBP38 since these phosphosites were up-regulated across all 4 replicates similar to the Rab GTPases upon PINK1 activation.

- In Fig. 3, the top panels from the MS assay show the extracted precursor ion signal for the pSer111 peptides. It would be useful for the reader if this figure or a corresponding Suppl Figure would present the comparable unphosphorylated extracted ion chromatograms of this and/or another unmodified peptide for comparison.

We believe that the Coomassie gel band data already presented in the panels of Figure 3 demonstrate that equivalent amounts of starting material was analysed for each of the conditions. For the information of the reviewer we attach to this cover letter Appendix Figures 1-3 of the analysis of the unphosphorylated extracted ion chromatograms for Rab8A, 8B and 13.

What mass tolerance was used for generating these MSI ion signals?

10ppm – this is now stated in the relevant Methods section.

What did the Coomassie gel look like and what gel region was excised for the analysis. Addition of these additional elements in a parallel Suppl figure would be informative.

As stated above, we have already shown Coomassie gel data included of the band that was excised for analysis in Figure 3. However, for the information of the reviewer we attach Appendix Figure 4, which shows the full Coomassie stained gels with Rab8A, 8B, and 13 bands that were excised marked with an asterisk.

Moreover, while a selection of the methods for this were currently presented in the legend, a more comprehensive description should be provided within the Methods section.

We have now added a comprehensive description in the Methods section.

- The authors are likely aware that LRRK2 is a Parkinson's Disease gene, kinase and RAB GTPase containing protein. Interestingly, this protein has been shown to undergo autophosphorylation on Ser1443, the residue equivalent to Ser111 in Rab8a/8b/13/1a. Is it possible that LRRK2 serves as the kinase downstream of PINK1 in the model systems used within this paper? Can exogenously expressed LRRK2 stimulate phosphorylation of Rab8a/8b/13/1a on Ser111 in the assay systems used in Figs 3-4?

We have addressed whether LRRK2 can mediate Ser111 phosphorylation in cells using 2

structurally distinct and selective LRRK2 inhibitors in cell based experiments. Under the assay conditions used we do not observe any reduction in Ser111 phosphorylation upon mitochondrial depolarisation suggesting that LRRK2 does not regulate this site in cells. This data has been included in Appendix Figure S9.

2nd Editorial Decision

07 September 2015

Thank you for submitting your revised manuscript for our consideration. Two of the original referees (see comments below) have now assessed it once more, and I am pleased to inform you that both of them are largely satisfied with your revisions and responses to the previous round of review. We shall therefore be happy to accept the paper for publication in The EMBO Journal, pending addressing of the two minor (editorial) concerns retained by referee 3. I am therefore returning the manuscript to you for a final round of revision, in order to allow you to incorporate the changes requested by this referee.

Once we will have received this final version and files, we should be able to swiftly proceed with formal acceptance and publication of the study. Should you have any further questions in this regard, please do not hesitate to get back to me.

REFEREE REPORTS

Referee #2:

The authors have addressed most of my concerns. I really wish that they had done more to address my point 4 - providing some in vivo evidence that Rab8 fails to be activated upon PINK activation. The Rab8-Rabin co-IP experiments provided in response to my comments don't directly address the point. Nonetheless, I do not want to hold the paper up on this issue as I am of the opinion that it is likely correct.

Referee #3:

The revised manuscript from the Trost and Muqit groups has satisfied the majority of my concerns in a satisfactory manner. Overall, this is an excellent biochemistry paper. The two outstanding issues relate to raw data availability and extracted ion chromatograms for Fig 3A-C. In response to the initial review, the authors deposited global phosphorylation data in the PRIDE repository under identifier PXD002127. Unfortunately the requested datafiles for the 'single protein analyses' in Fig 3 have not been provided. In their written response, the authors did kindly provide traces for the extracted ion chromatograms corresponding to total Rab protein levels stemming from these 'single protein analyses'. They imply that adding them as a Suppl Figure is unnecessary given that the Coomassie stained band is shown. Upon inspection though, the data appear to reveal differences in total Rab protein signal between samples that does not necessarily correlate to the intensities of the corresponding Coomassie stained bands. Given that the changes observed in the +WT PINK1 + CCCP treated condition for are nearly all or none, it seems unlikely (but not impossible) that these ratio data will lead the reader to the same conclusion as is reached in the text. That said, the most informative comparison between the six lanes in panels 3A-3C would be a ratio of phosphopeptide signal to unmodified peptide signal... which is possible from the data collected. It seems reasonable that readers be given access to these datafiles via the repository and the extracted ion traces as a Supplementary panel (arranged and labeled in the same order as the Fig. 3A)

2nd Revision - authors' response

14 September 2015

Major Points raised by Reviewer 3

The two outstanding issues relate to raw data availability and extracted ion chromatograms for Fig 3A-C. In response to the initial review, the authors deposited global phosphorylation data in the

PRIDE repository under identifier PXD002127. Unfortunately the requested datafiles for the 'single protein analyses' in Fig 3 have not been provided.

These datafiles have now been uploaded together with the global phosphorylation data in the PRIDE repository under the identifier PXD002127.

That said, the most informative comparison between the six lanes in panels 3A-3C would be a ratio of phosphopeptide signal to unmodified peptide signal which is possible from the data collected. It seems reasonable that readers be given access to these datafiles via the repository and the extracted ion traces as a Supplementary panel (arranged and labeled in the same order > as the Fig. 3A)

We disagree with the reviewer that the ratio of the phosphopeptide signal to the unmodified peptide signal would be the most informative comparison. Rather we have re-analysed the total protein intensities by quantifying the number of unique and razor peptides and this is depicted as non-normalised intensities for Rab8A, 8B and 13 in the revised Appendix Figs S5A, S6A and S7A respectively (in the same order as Fig 3A-C). We have then calculated the ratio of the phosphopeptide signal intensity to the whole protein signal intensities and this is expressed as the normalized relative phosphopeptide intensities and this is depicted for Rab8A, 8B and 13 in Appendix Figs S5D, S6D and S7D respectively. The resultant normalized data is consistent with the absolute data shown in Fig 3A-C for the phosphopeptide intensity.

Furthermore these data files have been deposited in the PRIDE repository under the identifier PXD002127.